# Assessing the causal role of epigenetic clocks in the development of multiple cancers: a Mendelian randomization study

Fernanda Morales Berstein[1,2]*, Daniel L McCartney[3], Ake T Lu[4], Konstantinos K Tsilidis[5,6], Emmanouil Bouras[6], Philip Haycock[1,2], Kimberley Burrows[1,2], Amanda I Phipps[7,8], Daniel D Buchanan[9], Iona Cheng[10], the PRACTICAL consortium, Richard M Martin[1,2,11], George Davey Smith[1,2], Caroline L Relton[1,2], Steve Horvath[4,12], Riccardo E Marioni[3], Tom G Richardson[1,2,13], Rebecca C Richmond[1,2]

[1]MRC Integrative Epidemiology Unit, University of Bristol, Bristol, United Kingdom; [2]Population Health Sciences, Bristol Medical School, Bristol, United Kingdom; [3]Centre for Genomic and Experimental Medicine, Institute of Genetics and Cancer, University of Edinburgh, Edinburgh, United Kingdom; [4]Department of Human Genetics, David Geffen School of Medicine, University of California, Los Angeles, Los Angeles, United States; [5]Department of Epidemiology and Biostatistics, School of Public Health, Imperial College London, London, United Kingdom; [6]Department of Hygiene and Epidemiology, School of Medicine, University of Ioannina, Ioannina, Greece; [7]Public Health Sciences Division, Fred Hutchinson Cancer Research Center, Seattle, United States; [8]Department of Epidemiology, School of Public Health, University of Washington, Seattle, United States; [9]Department of Clinical Pathology, Melbourne Medical School, University of Melbourne, Parkville, Australia; [10]Cancer Prevention Institute of California, Fremont, United States; [11]National Institute for Health Research (NIHR) Bristol Biomedical Research Centre, University Hospitals Bristol and Weston NHS Foundation Trust and the University of Bristol, Bristol, United Kingdom; [12]Department of Biostatistics, Fielding School of Public Health, University of California, Los Angeles, Los Angeles, United States; [13]Novo Nordisk Research Centre, Oxford, United Kingdom

*For correspondence:
dy20206@bristol.ac.uk

Group author details:
the PRACTICAL consortium See page 18

## Abstract

**Background:** Epigenetic clocks have been associated with cancer risk in several observational studies. Nevertheless, it is unclear whether they play a causal role in cancer risk or if they act as a non-causal biomarker.

**Methods:** We conducted a two-sample Mendelian randomization (MR) study to examine the genetically predicted effects of epigenetic age acceleration as measured by HannumAge (nine single-nucleotide polymorphisms (SNPs)), Horvath Intrinsic Age (24 SNPs), PhenoAge (11 SNPs), and GrimAge (4 SNPs) on multiple cancers (i.e. breast, prostate, colorectal, ovarian and lung cancer). We obtained genome-wide association data for biological ageing from a meta-analysis (N = 34,710), and for cancer from the UK Biobank (N cases = 2671–13,879; N controls = 173,493–372,016), FinnGen (N cases = 719–8401; N controls = 74,685–174,006) and several international cancer genetic consortia (N cases = 11,348–122,977; N controls = 15,861–105,974). Main analyses were performed using multiplicative random effects inverse variance weighted (IVW) MR. Individual study estimates were pooled using fixed effect meta-analysis. Sensitivity analyses included MR-Egger, weighted

median, weighted mode and Causal Analysis using Summary Effect Estimates (CAUSE) methods, which are robust to some of the assumptions of the IVW approach.

**Results:** Meta-analysed IVW MR findings suggested that higher GrimAge acceleration increased the risk of colorectal cancer (OR = 1.12 per year increase in GrimAge acceleration, 95% CI 1.04–1.20, p = 0.002). The direction of the genetically predicted effects was consistent across main and sensitivity MR analyses. Among subtypes, the genetically predicted effect of GrimAge acceleration was greater for colon cancer (IVW OR = 1.15, 95% CI 1.09–1.21, p = 0.006), than rectal cancer (IVW OR = 1.05, 95% CI 0.97–1.13, p = 0.24). Results were less consistent for associations between other epigenetic clocks and cancers.

**Conclusions:** GrimAge acceleration may increase the risk of colorectal cancer. Findings for other clocks and cancers were inconsistent. Further work is required to investigate the potential mechanisms underlying the results.

**Funding:** FMB was supported by a Wellcome Trust PhD studentship in Molecular, Genetic and Life-course Epidemiology (224982/Z/22/Z which is part of grant 218495/Z/19/Z). KKT was supported by a Cancer Research UK (C18281/A29019) programme grant (the Integrative Cancer Epidemiology Programme) and by the Hellenic Republic's Operational Programme 'Competitiveness, Entrepreneurship & Innovation' (OΠΣ 5047228). PH was supported by Cancer Research UK (C18281/A29019). RMM was supported by the NIHR Biomedical Research Centre at University Hospitals Bristol and Weston NHS Foundation Trust and the University of Bristol and by a Cancer Research UK (C18281/A29019) programme grant (the Integrative Cancer Epidemiology Programme). RMM is a National Institute for Health Research Senior Investigator (NIHR202411). The views expressed are those of the author(s) and not necessarily those of the NIHR or the Department of Health and Social Care. GDS and CLR were supported by the Medical Research Council (MC_UU_00011/1 and MC_UU_00011/5, respectively) and by a Cancer Research UK (C18281/A29019) programme grant (the Integrative Cancer Epidemiology Programme). REM was supported by an Alzheimer's Society project grant (AS-PG-19b-010) and NIH grant (U01 AG-18-018, PI: Steve Horvath). RCR is a de Pass Vice Chancellor's Research Fellow at the University of Bristol.

## Editor's evaluation

This paper is of broad interest to researchers seeking to disentangle the health impact of epigenetic age acceleration, and will provide a substantive empirical contribution to the literature. The authors were very meticulous in addressing all the concerns from the reviewers, which has further improved the paper.

## Introduction

DNA methylation (DNAm) at specific cytosine-phosphate-guanine (CpG) sites has been found to be strongly correlated with chronological age. Biological age, as predicted by DNAm patterns at specific CpG sites, may differ from chronological age on an individual basis. Observational evidence suggests that epigenetic age acceleration (i.e. when an individual's biological age is greater than their chronological age) may be associated with an increased risk of mortality and age-related diseases, including cancer (*Fransquet et al., 2019*).

Epigenetic clocks are heritable indicators of biological ageing derived from DNAm data. Each clock is based on DNAm levels measured at a different set of CpG sites, which capture distinctive features of epigenetic ageing (*Liu et al., 2020*). 'First-generation' epigenetic clocks, such as HannumAge (*Hannum et al., 2013*) and Intrinsic HorvathAge (*Horvath, 2013*), have been derived from DNAm levels at CpG sites found to be strongly associated with chronological age. HannumAge is trained on 71 age-related CpGs found in blood (*Hannum et al., 2013*), while Intrinsic HorvathAge is based on 353 age-related CpGs found in several human tissues and cell types, and is further adjusted for blood cell counts (*Horvath, 2013*). More recently, 'second-generation' epigenetic clocks, such as, PhenoAge (*Levine et al., 2018*) and GrimAge (*Lu et al., 2019a*), have been developed to predict age-related morbidity and mortality. PhenoAge incorporates data from 513 CpGs associated with mortality and nine clinical biomarkers (i.e. albumin, creatinine, serum glucose, C-reactive protein, lymphocyte percentage, mean corpuscular volume, red cell distribution width, alkaline phosphatase

**eLife digest** Have you noticed that some people seem to get older faster than others? Scientists have previously found that a chemical tag on DNA known as DNA methylation can be used to predict an individual's chronological age. However, age predicted using DNA methylation (also known as biological or epigenetic age) does not always perfectly correspond to chronological age. Indeed, some people's biological age is higher than their years, while other people's is lower.

When an individual's biological age is higher than their chronological age, they are said to be experiencing 'epigenetic age acceleration'. This type of accelerated ageing, which can be measured with 'epigenetic clocks' based on DNA methylation, has been associated with several adverse health outcomes, including cancer. This means that epigenetic clocks may improve our ability to predict cancer risk and detect cancer early. However, it is still unclear whether accelerated biological ageing causes cancer, or whether it simply correlates with the disease.

Morales-Berstein et al. wanted to investigate whether epigenetic age acceleration, as measured by epigenetic clocks, plays a role in the development of several cancers. To do so, they used an approach known as Mendelian randomization. Using genetic variants as natural experiments, they studied the effect of different measures of epigenetic age acceleration on cancer risk.

Their work focused on five types of cancer: breast, colorectal, prostate, ovarian and lung cancer. They used genetic association data from people of European ancestry to determine whether genetic variants that are strongly associated with accelerated ageing are also strongly associated with cancer. The results showed that one of the DNA methylation markers used as an estimate of biological ageing could be directly related to the risk of developing colorectal cancer.

This work provides new insights into the relationship between markers of biological ageing and cancer. Similar relationships should also be studied in other groups of people and for other cancer sites. The results suggest that reversing biological ageing by altering DNA methylation could prevent or delay the development of colorectal cancer.

and leukocyte count) (*Levine et al., 2018*), and GrimAge includes data from 1,030 CpGs associated with smoking pack-years and seven plasma proteins (i.e. cystatin C, leptin, tissue inhibitor metalloproteinases 1, adrenomedullin, beta-2-microglobulin, growth differentiation factor 15, and plasminogen activation inhibitor 1 (PAI-1)) (*Lu et al., 2019a*). Due to differences in their composition, HannumAge and Intrinsic HorvathAge are better predictors of chronological age (*Hannum et al., 2013*; *Horvath, 2013*), while PhenoAge and GrimAge stand out for their ability to predict health and lifespan (*Levine et al., 2018*; *Lu et al., 2019a*; *McCrory et al., 2021*).

Several studies suggest that HannumAge, Intrinsic HorvathAge, PhenoAge and GrimAge acceleration are associated with cancer risk (*Levine et al., 2018*; *Ambatipudi et al., 2017*; *Levine et al., 2015*; *Dugue et al., 2021*; *Kresovich et al., 2019b*; *Kresovich et al., 2019a*; *Zheng et al., 2016*). In contrast, others indicate that evidence in support of this claim is weak or non existent (*Dugué et al., 2018*; *Hillary et al., 2020*; *Durso et al., 2017*; *Wang et al., 2021*). This lack of consensus could be explained by biases that often affect observational research, such as reverse causation (e.g. cancer influencing the epigenome and not the other way around) and residual confounding (e.g. unmeasured, or imprecisely measured confounders of the association between epigenetic age acceleration and cancer) (*Relton and Davey Smith, 2012*).

The strength of the associations between epigenetic age acceleration and different cancers has also been found to vary across epigenetic clocks. For instance, positive associations between epigenetic age acceleration and colorectal cancer seem to be much stronger when biological age is estimated using second-generation clocks (i.e. PhenoAge and GrimAge) (*Dugue et al., 2021*) rather than first-generation clocks (i.e. HannumAge and Intrinsic HorvathAge) (*Dugué et al., 2018*; *Durso et al., 2017*). Lack of consensus across epigenetic clocks could be explained by differences in their algorithms (which may reflect different mechanisms of biological ageing), as well as heterogeneity in study designs (*Fransquet et al., 2019*). Furthermore, even if there were a consensus, it would still be unclear whether age-related DNA methylation plays a causal role in cancer risk or if it merely acts as a non-causal prognostic biomarker.

Mendelian randomization (MR), a method that uses genetic variants as instrumental variables to infer causality between a modifiable exposure and an outcome, is less likely to be affected by residual confounding and reverse causation than traditional observational methods (*Davey Smith and Ebrahim, 2003*). A recent genome-wide association study (GWAS) meta-analysis has revealed 137 genetic loci associated with epigenetic age acceleration (as measured by six epigenetic biomarkers) that may be used within an MR framework (*McCartney et al., 2021*).

*McCartney et al., 2021* used IVW MR, MR-Egger, weighted median and weighted mode methods to explore the genetically predicted effects of HannumAge, Intrinsic HorvathAge, PhenoAge and GrimAge acceleration on breast, ovarian, and lung cancer. Here, we extend this analysis to include colorectal and prostate cancer (two of the most common cancers worldwide *Sung et al., 2021*) and use additional methods and datasets to verify the robustness of our findings.

The aim of this two-sample MR study was to examine the genetically predicted effects of epigenetic age acceleration (as measured by HannumAge *Hannum et al., 2013*, Horvath Intrinsic Age *Horvath, 2013*, PhenoAge *Levine et al., 2018* and GrimAge *Lu et al., 2019a*) on multiple cancers (i.e., breast, prostate, colorectal, ovarian and lung cancer) using summary genetic association data from (1) McCartney et al. (N = 34,710) (*McCartney et al., 2021*), (2) the UK Biobank (N cases = 2671–13,879; N controls = 173,493–372,016), (3) FinnGen (N cases = 719–8401; N controls = 74,685–174,006) and (4) several international cancer genetic consortia (N cases = 11,348–122,977; N controls = 15,861–105,974).

## Materials and methods
### Reporting guidelines
This study has been reported according to the STROBE-MR guidelines (*Skrivankova et al., 2021*; *Supplementary file 2*).

**Table 1.** Numbers of overall cancer cases and controls by data source.

| Cancer type | Source | N cases (%)[*] | N controls |
|---|---|---|---|
| Breast | BCAC | 122,977 (53.7%) | 105,974 |
| | UK Biobank | 13,879 (6.5%) | 198,523 |
| | FinnGen | 8401 (7.8%) | 99,321 |
| Ovarian | OCAC | 25,509 (38.4%) | 40,941 |
| | UK Biobank | 1218 (0.6%) | 198,523 |
| | FinnGen | 719 (0.7%) | 99,321 |
| Prostate | PRACTICAL | 79,148 (56.4%) | 61,106 |
| | UK Biobank | 9132 (5.0%) | 173,493 |
| | FinnGen | 6311 (7.8%) | 74,685 |
| Lung | ILCCO | 11,348 (41.7%) | 15,861 |
| | UK Biobank | 2671 (0.7%) | 372,016 |
| | FinnGen | 1681 (1.0%) | 173,933 |
| Colorectal | GECCO | 58,131 (46.3%) | 67,347 |
| | UK Biobank | 5657 (1.5%) | 372,016 |
| | FinnGen | 3022 (1.7%) | 174,006 |

[*]Percentage (%) of cases within each study source was calculated using the following formula: 100 * N cases / (N cases + N controls).

BCAC = Breast Cancer Association Consortium. OCAC = Ovarian Cancer Association Consortium. PRACTICAL = Prostate Cancer Association Group to Investigate Cancer Associated Alterations in the Genome. ILCCO = International Lung Cancer Consortium. GECCO = Genetics and Epidemiology of Colorectal Cancer Consortium.

## Genetic instruments for epigenetic age acceleration

We obtained summary genetic association estimates for epigenetic age acceleration measures of HannumAge (*Hannum et al., 2013*), Intrinsic HorvathAge (*Horvath, 2013*), PhenoAge (*Levine et al., 2018*), and GrimAge (*Lu et al., 2019a*) from a recent GWAS meta-analysis of biological ageing (*McCartney et al., 2021*), which included 34,710 participants of European ancestry. Across the 28 European ancestry studies considered in the analysis, 57.3% of participants were female. A detailed description of the methods that were used can be found in the publication by *McCartney et al., 2021*. In short, the Horvath epigenetic age calculator software (https://dnamage.genetics.ucla.edu) or standalone scripts were used to calculate age adjusted DNAm estimates. Outlier samples with clock methylation estimates of +/−5 s.d. from the mean were excluded from further analysis. SNPs were genotyped and imputed independently for each cohort included in the meta-analysis. Genotypes were imputed using either the HRC or the 1000 Genomes Project Phase 3 reference panels in all cohorts but the Sister Study (which did not have imputed data at the time of analysis) and the Genetics of Lipid Lowering Drugs and Diet Network Study (which used whole-genome sequencing data). GWAS summary statistics were obtained in each cohort using additive linear models adjusted for sex and genetic principal components, and they were later processed and harmonised using the 'EasyQC' R package. Fixed effect meta-analyses were performed using the METAL software (*Willer et al., 2010*).

We used the clump_data function in the 'TwoSampleMR' R package to select GWAS-significant SNPs ($P < 5 \times 10^{-8}$) for each epigenetic age acceleration measure and perform linkage disequilibrium (LD) clumping ($r^2 < 0.001$) using the European reference panel from the 1000 Genomes Project Phase 3 v5.

We identified 9 independent SNPs for HannumAge, 24 for Intrinsic HorvathAge, 11 for PhenoAge and 4 for GrimAge (*Supplementary file 1 — Table 1*). The proportions of trait variance explained by genetic instruments ($R^2$) and instrument strength (F-statistic) were calculated using the following formulae: $R^2 = (2\beta^2 \times MAF \times (1-MAF))/(2\beta^2 \times MAF \times (1-MAF) + 2 N \times MAF \times (1-MAF) \times SE^2)$ and $F = (R^2 \times (N-2))/(1-R^2)$ (where MAF = effect allele frequency, $\beta$ = effect estimate of the SNP in the exposure GWAS, SE = standard error, N = sample size) (*Palmer et al., 2012*). The genetic instruments for HannumAge, Intrinsic HorvathAge, PhenoAge and GrimAge acceleration explained 1.48%, 4.41%, 1.86%, and 0.47% of the trait variance, respectively. All the selected SNPs had F-statistics greater than 10 (HannumAge median 38 and range 31–99, Intrinsic HorvathAge median 47 and range 31–240, PhenoAge median 45 and range 32–89, GrimAge median 36 and range 31–45).

## Genetic Association Data sources for cancer outcomes

We obtained summary-level genetic association data for cancer outcomes from the UK Biobank, FinnGen and several international cancer genetic consortia: the Breast Cancer Association Consortium (BCAC), the Ovarian Cancer Association Consortium (OCAC), the Consortium of Investigators of Modifiers of BRCA1/2 (CIMBA), the Prostate Cancer Association Group to Investigate Cancer Associated Alterations in the Genome (PRACTICAL), the International Lung Cancer Consortium (ILCCO) and the Genetics and Epidemiology of Colorectal Cancer Consortium (GECCO) (*Table 1*). Further details of the studies and the data obtained are described in **Appendix 1**.

We extracted genetic association data for the selected SNPs from each cancer GWAS (for breast, prostate, colorectal, ovarian and lung cancers). LD proxies ($r^2 > 0.8$) were used when the SNPs of interest were missing from the cancer GWAS dataset. The proxies were located using the MR-Base platform, which calculates LD using the European subset of individuals from the 1000 Genomes Project reference panel as above (*Hemani et al., 2018*). The 'LDlinkR' R package version 1.1.2 was used to find proxies for cancer data that were not included in the MR-Base platform. The exposure and outcome datasets were then harmonised to ensure the genetic associations reflect the same effect allele. Palindromic SNPs with minor allele frequencies (MAF) <0.3 were aligned, while those with MAF ≥0.3 or mismatching strands were excluded.

## Power calculations

Statistical power was calculated using an online calculator for MR available at: https://shiny.cnsgenomics.com/mRnd/. Calculations were performed separately for each clock-cancer combination. They were based on a type one error rate of 0.05, the proportion of phenotypic variance explained by genetic variants ($R^2$) for each measure of epigenetic age acceleration, and the total number of cases

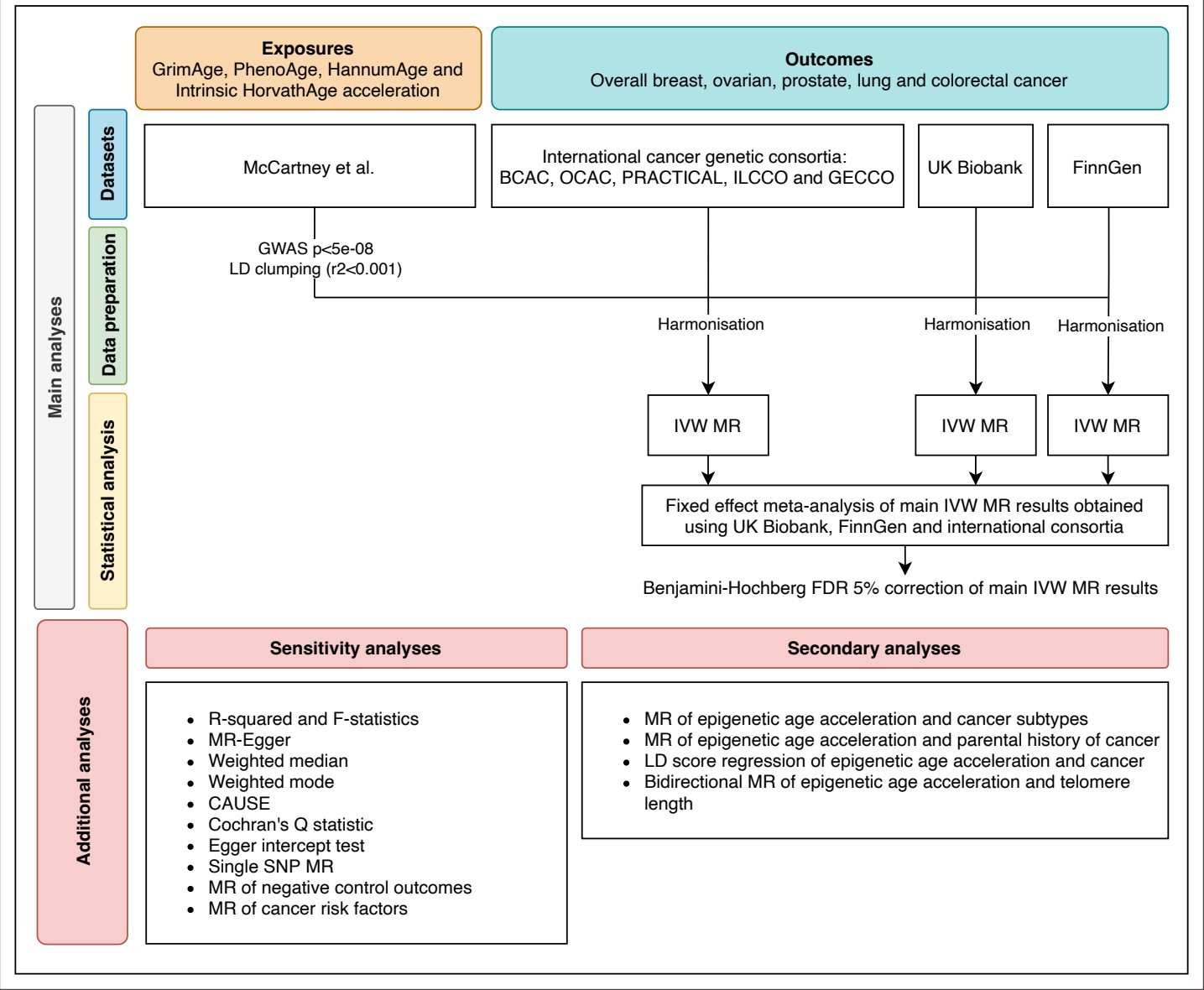

**Figure 1.** Flowchart summarising study methods. Abbreviations: BCAC, Breast Cancer Association Consortium; OCAC, Ovarian Cancer Association Consortium; PRACTICAL, Prostate Cancer Association Group to Investigate Cancer Associated Alterations in the Genome; ILCCO, International Lung Cancer Consortium; GECCO, Genetics and Epidemiology of Colorectal Cancer Consortium; LD, linkage disequilibrium; IVW, inverse variance weighted; MR, Mendelian randomization; FDR, false discovery rate; GWAS, genome-wide association study; CAUSE, Causal Analysis Using Summary Effect estimates, SNP, single-nucleotide polymorphism.

and controls included in the meta-analysis for each cancer. Across combinations of the four epigenetic clock acceleration and five cancer measures, we had 80% power to detect ORs as small as 1.04–1.39 (*Supplementary file 1* — Table s2).

## Statistical analysis

We estimated the genetically predicted effects of epigenetic age acceleration (as measured by HannumAge *Hannum et al., 2013*, Horvath Intrinsic Age *Horvath, 2013*, PhenoAge *Levine et al., 2018* and GrimAge *Lu et al., 2019a*) on multiple cancers (i.e. breast, prostate, colorectal, ovarian, and lung cancer) using a two-sample MR framework (*Figure 1*).

## Main analyses

Main analyses were performed using multiplicative random effects inverse variance weighted (IVW) MR, a method that combines the genetically predicted effect of epigenetic age acceleration on cancer across genetic variants (*Burgess et al., 2013*). This is the default IVW MR method in the 'TwoSampleMR' R package, as it accounts for excess heterogeneity across SNP-specific estimates (as opposed to the fixed effect IVW method) and it does not affect the relative weights of individual SNP estimates (in contrast to the additive random effects IVW method) (*Bowden et al., 2017*).

We used fixed effect meta-analysis to pool results across studies (i.e. UK Biobank, FinnGen and international consortia). For colorectal cancer, we only pooled FinnGen and GECCO estimates, since UK Biobank participants were already included in GECCO. $I^2$ statistics and their corresponding confidence intervals were used to estimate heterogeneity across study estimates (*von Hippel, 2015*). A Benjamini-Hochberg false discovery rate (FDR) < 5% was used to correct the pooled main IVW results for multiple testing (*Benjamini and Hochberg, 1995*). This correction was applied considering a total of 20 independent statistical tests (4 clocks x 5 cancers = 20).

## Sensitivity analyses

MR assumes genetic instruments for epigenetic age acceleration are (1) associated with epigenetic age acceleration (relevance assumption), (2) independent of confounders of the association between the instruments and cancer (independence assumption), and (3) only associated with cancer through their effect on epigenetic age acceleration (exclusion restriction assumption) (*Didelez and Sheehan, 2007*; *Davies et al., 2018*).

As a sensitivity analysis and to test for potential violations of the relevance assumption, we calculated F-statistics and the $R^2$ for each measure of epigenetic age acceleration (*Burgess and Thompson, 2011*). Other sensitivity analyses included MR-Egger (*Bowden et al., 2015*), weighted median (*Bowden et al., 2016*) and weighted mode (*Hartwig et al., 2017*) methods, which are robust to some of the assumptions of the IVW approach (described in **Appendix 1**). These results were also pooled across studies, as explained above. Consistency across different MR methods would suggest that it is less likely that the independence and exclusion restriction assumptions are violated.

We further assessed the validity of the independence assumption by conducting MR analyses using negative control outcomes (i.e. skin colour, ease of skin tanning). Evidence of causality between our genetic instruments for epigenetic age acceleration and these negative control outcomes would suggest potential bias due to population stratification that has not been fully accounted for through adjustments in the GWAS (*Sanderson et al., 2021*). We also assessed the genetically predicted effect of epigenetic age acceleration on cancer risk factors (i.e. body mass index, waist circumference, pack years of smoking, time spent doing vigorous physical activity, age completed full time education, years of schooling, and alcohol intake frequency) to detect potential violations of the exclusion restriction assumption. GWAS data for negative control outcomes and cancer risk factors were obtained using the University of Bristol's IEU OpenGWAS API (for more details, see **Appendix 1**).

Where associations between genetically predicted epigenetic age acceleration and cancer were identified, we additionally performed single-SNP two-sample MR analysis to assess whether the effects were likely to be driven by a single SNP. We used the METAL software (*Willer et al., 2010*) to conduct a GWAS meta-analysis of cancer genetic association data obtained from the UK Biobank, FinnGen and international cancer genetic consortia. We then used these meta-analysed summary statistics in two-sample MR analyses. Scatter plots showing the effects of genetic instruments on epigenetic clock acceleration against their effects on cancer were created using the 'TwoSampleMR' R package. Additionally, Cochran's Q statistics were used to quantify global heterogeneity across SNP-specific MR estimates (*Bowden et al., 2019*) and MR-Egger intercept tests were performed to detect horizontal pleiotropy (*Bowden et al., 2015*).

We also used Causal Analysis using Summary Effect Estimates (CAUSE) (*Morrison et al., 2020*), a method that uses genome-wide summary statistics to disentangle causality (i.e. SNPs are associated with cancer through their effect on epigenetic age acceleration) from correlated horizontal pleiotropy (i.e. SNPs are associated with epigenetic age acceleration and cancer through a shared heritable factor), while taking into account uncorrelated horizontal pleiotropy (i.e. SNPs are associated with epigenetic age acceleration through separate mechanisms). It uses Bayesian modelling to assess

whether the sharing model (i.e. model that fixes the causal effect at zero) fits the data at least as well as the causal model (i.e. model that allows a causal effect different from zero).

## Secondary analyses

As a secondary analysis, we conducted two-sample MR of epigenetic age acceleration and cancer subtypes (i.e. breast cancer: ER+, ER-, triple negative, luminal B/HER2-negative-like, HER2-enriched-like, luminal A-like, luminal B-like, BRCA1 and BRCA2; ovarian cancer: high-grade serous, low-grade serous, invasive mucinous, clear cell, endometrioid, BRCA1 and BRCA2; prostate cancer: advanced, advanced [vs non-advanced], early onset, high risk [vs low risk], and high risk [vs low and intermediate risk]; lung cancer: adenocarcinoma and squamous cell; colorectal cancer: colon-specific, proximal colon-specific, distal colon-specific, rectal-specific, male and female) (**Appendix 1**).

We also performed two-sample MR analyses of epigenetic age acceleration and parental history of cancer in the UK Biobank for breast, prostate, lung and bowel cancer (**Appendix 1**). Data on parental history of ovarian cancer were not available in UK Biobank. Family history data correlate with combined hospital record and questionnaire data and it has been suggested that they provide better power to detect GWAS-significant associations for some phenotypes in the UK Biobank (*DeBoever et al., 2020*). Therefore, we expected these results to be consistent with those obtained in the main analyses.

MR results were reported as the odds ratio (OR) of site-specific cancer per one year increase in genetically predicted epigenetic age acceleration. These did not require any scale transformations, as the GWAS of biological ageing (*McCartney et al., 2021*) reported epigenetic age acceleration in years.

LD Score regression (*Bulik-Sullivan et al., 2015*) was used to identify genome-wide genetic correlations between epigenetic age acceleration and cancer. Genetic correlations were estimated using full GWAS summary statistics for the epigenetic clocks and cancer, as well as the 1000 Genomes Project European LD reference panel. Traits with mean heritability chi-square values < 1.02 were excluded from the analyses.

Finally, bidirectional MR analyses were conducted to assess the causality and directionality of the link between epigenetic clock acceleration and telomere length, another measure of biological ageing that has been shown to influence cancer risk in prior MR studies (*Telomeres Mendelian Randomization Collaboration et al., 2017*; *Gao et al., 2020*; *Kuo et al., 2019*). The MR Steiger test of directionality was used to confirm the assumption that the exposure causes the outcome is valid (*Hemani et al., 2017*). We also corroborated our findings by rerunning the analyses using data that had undergone Steiger filtering to remove SNPs that explained more variance in the outcome than in the risk factor. Genetic association data for measured telomere length were obtained from *Codd et al., 2021*, the largest GWAS of telomere length available through the OpenGWAS API at the time of analysis (N = 472,174, for more details, see **Appendix 1**).

All MR analyses were performed using R software version 4.0.2. Two sample MR analyses were conducted using the 'TwoSampleMR' package version 0.5.5. Meta-analyses of IVW results were performed using the 'meta' package version 4.18. GWAS meta-analyses used to perform single-SNP MR analyses were done using the METAL software (*Willer et al., 2010*). CAUSE analyses were conducted using the 'cause' package version 1.2.0. Forest plots were created using the 'ggforestplot' package version 0.1.0. LD Scores were computed using the 'ldsc' command line tool version 1.0.1. The code used in this study is available at: https://github.com/fernandam93/epiclocks_cancer.

## Results

### Breast cancer

We did not find strong evidence of causality between epigenetic age acceleration and breast cancer (GrimAge IVW OR = 0.98, 95% CI 0.95–1.00, p = 0.08; PhenoAge IVW OR = 0.99, 95% CI 0.98–1.01, p = 0.23; HannumAge IVW OR = 0.99, 95% CI 0.97–1.02, p = 0.63; and Intrinsic HorvathAge IVW OR = 0.99, 95% CI 0.98–1.00, p = 0.13) (*Figure 2*, *Appendix 2—figure 1*, *Appendix 2—figure 2*, *Appendix 2—figure 3*, *Appendix 2—figure 4*, *Appendix 2—figure 5*, *Appendix 2—figure 6*, *Appendix 2—figure 7*, *Appendix 2—figure 8*, *Supplementary file 1* — Table s3, *Supplementary file 1* — Table s4, *Supplementary file 1* — Table s5, *Supplementary file 1* — Table s6).

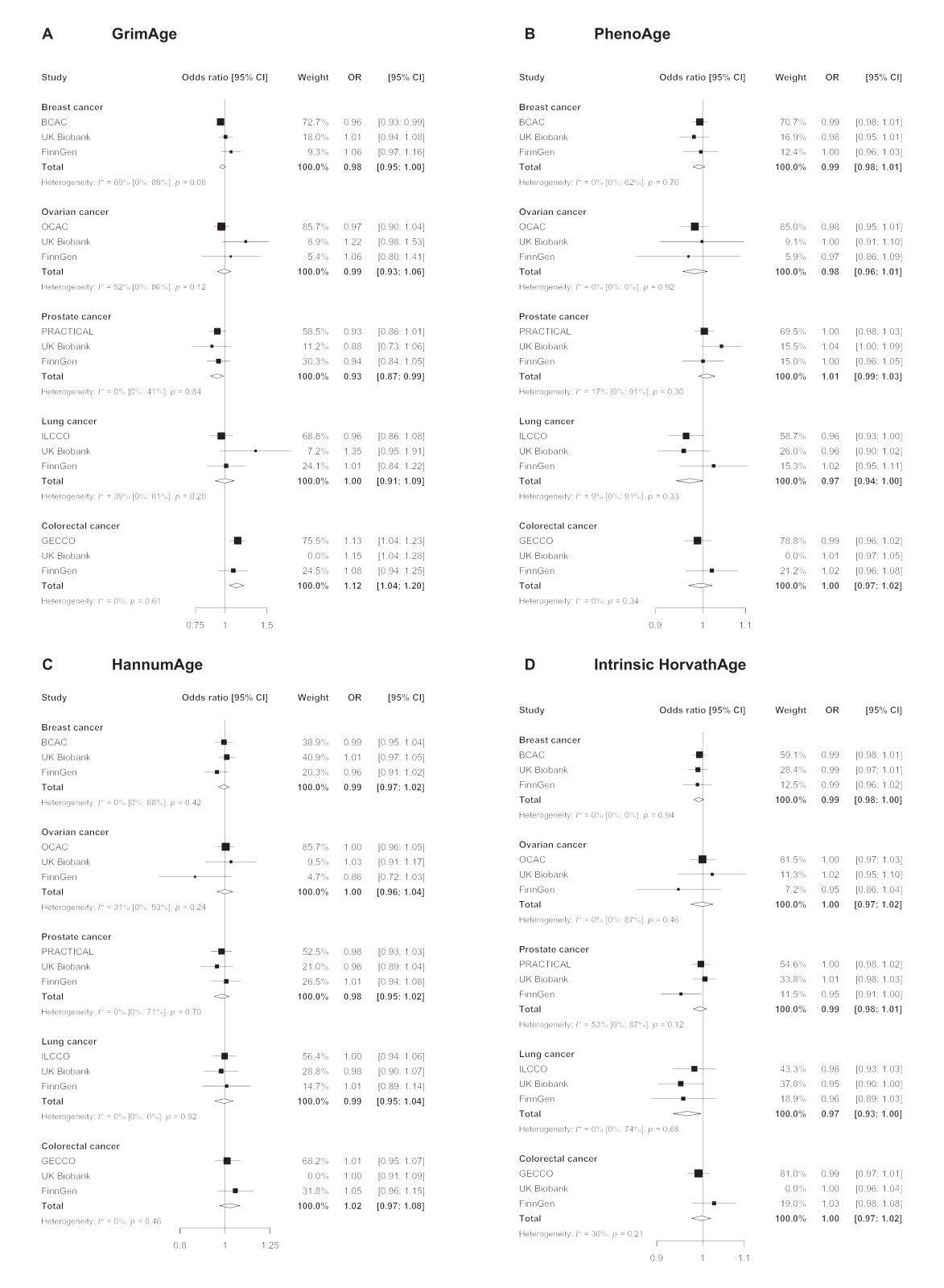

**Figure 2.** Fixed effect meta-analysis of inverse-variance weighted Mendelian randomization estimates for genetically predicted effects of epigenetic age acceleration on multiple cancers. Odds ratios and 95% confidence intervals are reported per 1 year increase in (A) GrimAge acceleration, (B) PhenoAge acceleration, (C) HannumAge acceleration and (D) Intrinsic HorvathAge acceleration. GrimAge, PhenoAge, HannumAge and Intrinsic HorvathAge acceleration were instrumented by 4, 11, 9, and 24 genetic variants, respectively. All meta-analysis estimates were calculated using data from UK Biobank, FinnGen and international consortia, except for colorectal cancer estimates, which exclude UK Biobank data to avoid double counting.

## Ovarian cancer

There was also limited evidence of causality between epigenetic age acceleration and ovarian cancer (GrimAge IVW OR = 0.99, 95% CI 0.93–1.06, p = 0.78; PhenoAge IVW OR = 0.98, 95% CI 0.96–1.01, p = 0.24; HannumAge IVW OR = 1.00, 95% CI 0.96–1.04, p = 0.95; and Intrinsic HorvathAge IVW OR = 1.00, 95% CI 0.97–1.02, p = 0.89) (*Figure 2*, *Appendix 2—figure 1*, *Appendix 2—figure 2*, *Appendix 2—figure 3*, *Appendix 2—figure 4*, *Appendix 2—figure 5*, *Appendix 2—figure 6*, *Appendix 2—figure 7*, *Appendix 2—figure 8*, *Supplementary file 1* — Table s3, *Supplementary file 1* — Table s4, *Supplementary file 1* — Table s5, *Supplementary file 1* — Table s6).

## Prostate cancer

Meta-analysed IVW MR findings suggested that genetically predicted GrimAge acceleration decreased the risk of prostate cancer (OR = 0.93 per year increase in GrimAge acceleration, 95% CI 0.87–0.99, p = 0.02) (*Figure 2*, *Supplementary file 1* — Table s3, *Supplementary file 1* — Table s4, *Supplementary file 1* — Table s5, *Supplementary file 1* — Table s6). Although the direction of the genetically predicted effect was consistent across main and sensitivity MR analyses (i.e. MR-Egger, weighted median and weighted mode) (*Appendix 2—figure 1*, *Supplementary file 1* — Table s3, *Supplementary file 1* — Table s4, *Supplementary file 1* — Table s5, *Supplementary file 1* — Table s6), the main IVW result for GrimAge and prostate cancer did not withstand multiple testing correction (FDR p = 0.16) (*Supplementary file 1* — Table s6).

We did not find consistent evidence of causality between other measures of epigenetic age acceleration and prostate cancer (PhenoAge IVW OR = 1.01, 95% CI 0.99–1.03, p = 0.31; HannumAge IVW OR = 0.98, 95% CI 0.95–1.02, p = 0.39; and Intrinsic HorvathAge IVW OR = 0.99, 95% CI 0.98–1.01, p = 0.42) (*Figure 2*, *Appendix 2—figure 1*, *Appendix 2—figure 2*, *Appendix 2—figure 3*, *Appendix 2—figure 4*, *Appendix 2—figure 5*, *Appendix 2—figure 6*, *Appendix 2—figure 7*, *Appendix 2—figure 8*, *Supplementary file 1* — Table s3, *Supplementary file 1* — Table s4, *Supplementary file 1* — Table s5, *Supplementary file 1* — Table s6).

## Lung cancer

Meta-analysed IVW MR findings suggested that genetically predicted Intrinsic HorvathAge acceleration decreased the risk of lung cancer (OR = 0.97 per year increase in Intrinsic HorvathAge acceleration, 95% CI 0.93–1.00, p = 0.03) (*Figure 2*, *Supplementary file 1* — Table s3, *Supplementary file 1* — Table s4, *Supplementary file 1* — Table s5, *Supplementary file 1* — Table s6). However, these results did not survive multiple testing correction (FDR p = 0.21) and were not strongly supported by sensitivity analyses (*Appendix 2—figure 1*, *Supplementary file 1* — Table s3, *Supplementary file 1* — Table s4, *Supplementary file 1* — Table s5, *Supplementary file 1* — Table s6).

We did not find evidence of causality between other measures of epigenetic age acceleration and lung cancer (GrimAge IVW OR = 1.00, 95% CI 0.91–1.09, p = 0.96; PhenoAge IVW OR = 0.97, 95% CI 0.94–1.00, p = 0.06; and HannumAge IVW OR = 0.99, 95% CI 0.95–1.04, p = 0.82) (*Figure 2*, *Appendix 2—figure 1*, *Appendix 2—figure 2*, *Appendix 2—figure 3*, *Appendix 2—figure 4*, *Appendix 2—figure 5*, *Appendix 2—figure 6*, *Appendix 2—figure 7*, *Appendix 2—figure 8*, *Supplementary file 1* — Table s3, *Supplementary file 1* — Table s4, *Supplementary file 1* — Table s5, *Supplementary file 1* — Table s6).

## Colorectal cancer

Meta-analysed IVW MR findings suggested that genetically predicted GrimAge acceleration increased the risk of colorectal cancer (OR = 1.12 per year increase in GrimAge acceleration, 95% CI 1.04–1.20, p = 0.002) (*Figure 2*, *Supplementary file 1* — Table s3, *Supplementary file 1* — Table s5, *Supplementary file 1* — Table s6). These results survived multiple testing correction (FDR p = 0.04) and there was little evidence of heterogeneity across FinnGen and GECCO estimates ($I^2$ = 0%, 95% CI 'NA', p = 0.61). Additionally, the direction of the genetically predicted effect was consistent across main and sensitivity MR analyses (i.e. MR-Egger, weighted median, and weighted mode) (*Figure 3*, *Supplementary file 1* — Table s3, *Supplementary file 1* — Table s5, *Supplementary file 1* — Table s6) and results were consistent when using UK Biobank data alone (IVW OR = 1.15, 95% CI 1.04–1.28, p = 0.007) (*Figure 2*, *Supplementary file 1* — Table s4).

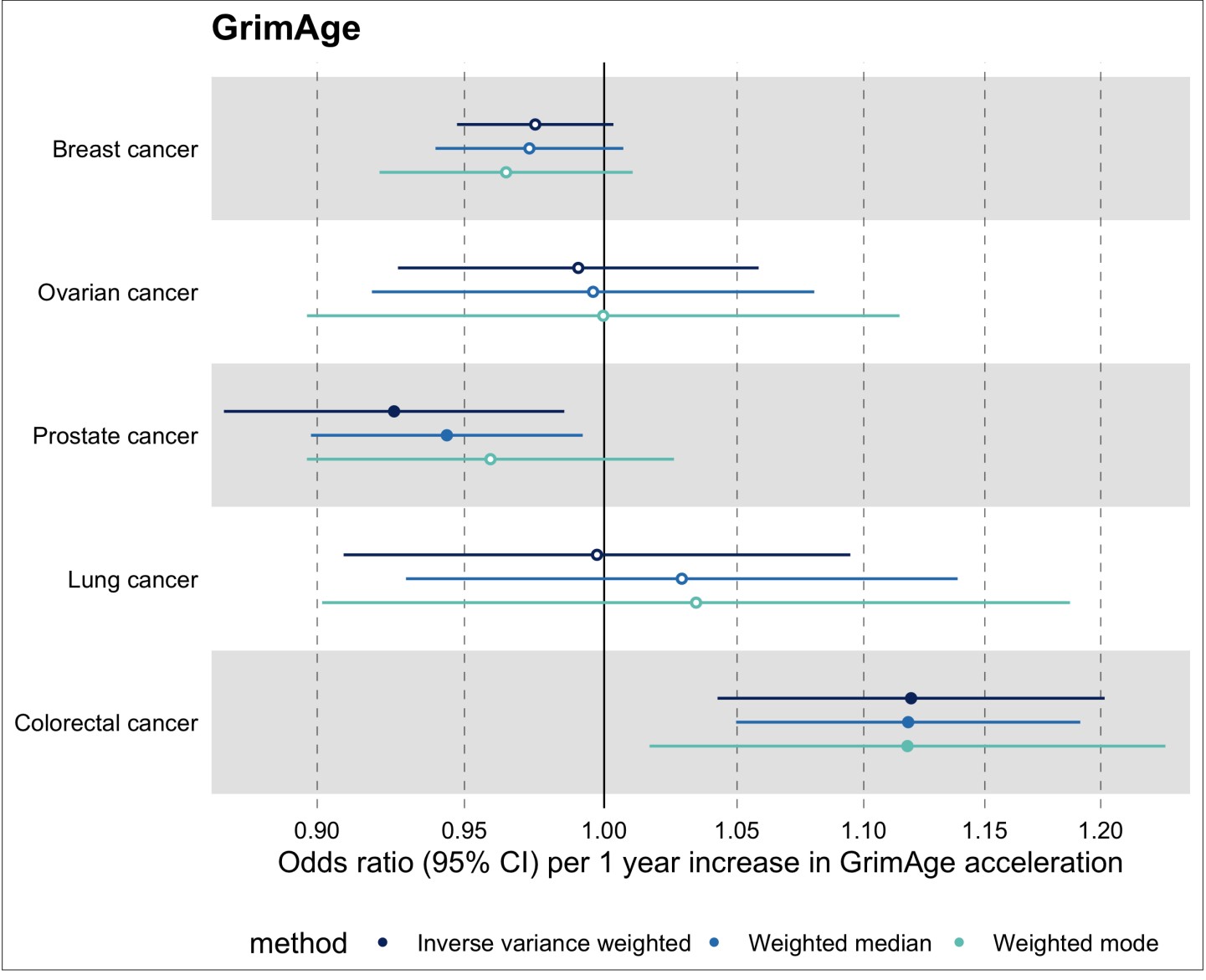

**Figure 3.** Fixed effect meta-analysis of Mendelian randomization estimates for genetically predicted effects of GrimAge acceleration on multiple cancers. Odds ratios and 95% confidence intervals are reported per 1 year increase in GrimAge acceleration. GrimAge acceleration was instrumented by four genetic variants. Results were obtained using inverse variance weighted MR (dark blue), weighted median (sky blue) and weighted mode (turquoise) methods. All meta-analysis estimates were calculated using data from UK Biobank, FinnGen and international consortia, except for colorectal cancer estimates, which exclude UK Biobank data to avoid double counting.

We did not find evidence of residual population stratification in MR analyses using negative control outcomes (*Appendix 2—figure 9*, *Supplementary file 1* — Table s7), nor did we find evidence of horizontal pleiotropy via potential colorectal cancer risk factors (*Appendix 2—figure 10*, *Supplementary file 1* — Table s8).

Single-SNP analysis revealed that the effect was not driven by a single SNP (*Supplementary file 1* — Table s9). *Figure 4* shows the effect of genetic instruments on GrimAge acceleration against their effect on colorectal cancer. Moreover, there was no detectable evidence of uncorrelated horizontal pleiotropy (MR-Egger intercept = –0.13, 95% CI –0.33–0.07, p = 0.33), or heterogeneity across individual SNP estimates (Cochran's Q 7.12, p = 0.07) (*Supplementary file 1* — Table s10). We further explored the genetically predicted effect of GrimAge on colorectal cancer using GECCO data only and found no evidence against bias due to correlated pleiotropy (CAUSE OR = 1.00, 95% credible intervals 0.96–1.04, p = 1.00; shared $q$ = 4%, 95% credible intervals 0–24%) (*Appendix 2—figure 11*).

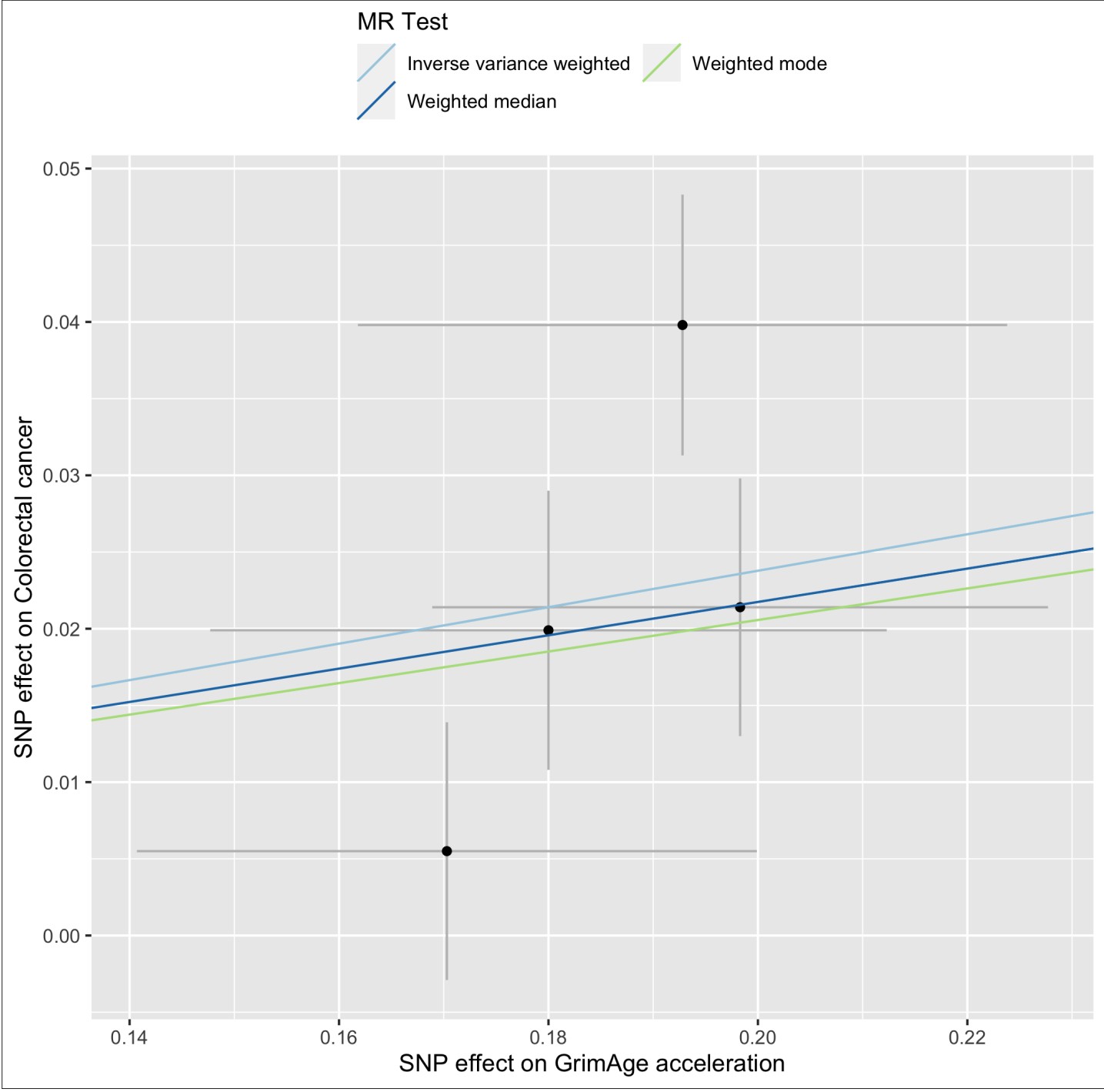

**Figure 4.** Scatter plot showing the effect of genetic instruments on GrimAge acceleration against their effect on colorectal cancer. FinnGen and Genetics and Epidemiology of Colorectal Cancer (GECCO) genome-wide association estimates for colorectal cancer were meta-analysed using the METAL software. UK Biobank estimates were not included in the meta-analysis to avoid double counting participants included in the GECCO consortium. Results were obtained using inverse variance weighted MR (light blue), weighted median (dark blue) and weighted mode (light green) methods.

Among subtypes, we found strong evidence for a causal relationship between GrimAge accelera-tion and colon cancer (IVW OR = 1.15, 95% CI 1.09–1.21, p = 0.006). In contrast, we did not find such evidence for rectal cancer (IVW OR = 1.05, 95% CI 0.97–1.13, p = 0.24). After further stratification, the magnitude of the genetically predicted effect of GrimAge acceleration on colon cancer was the

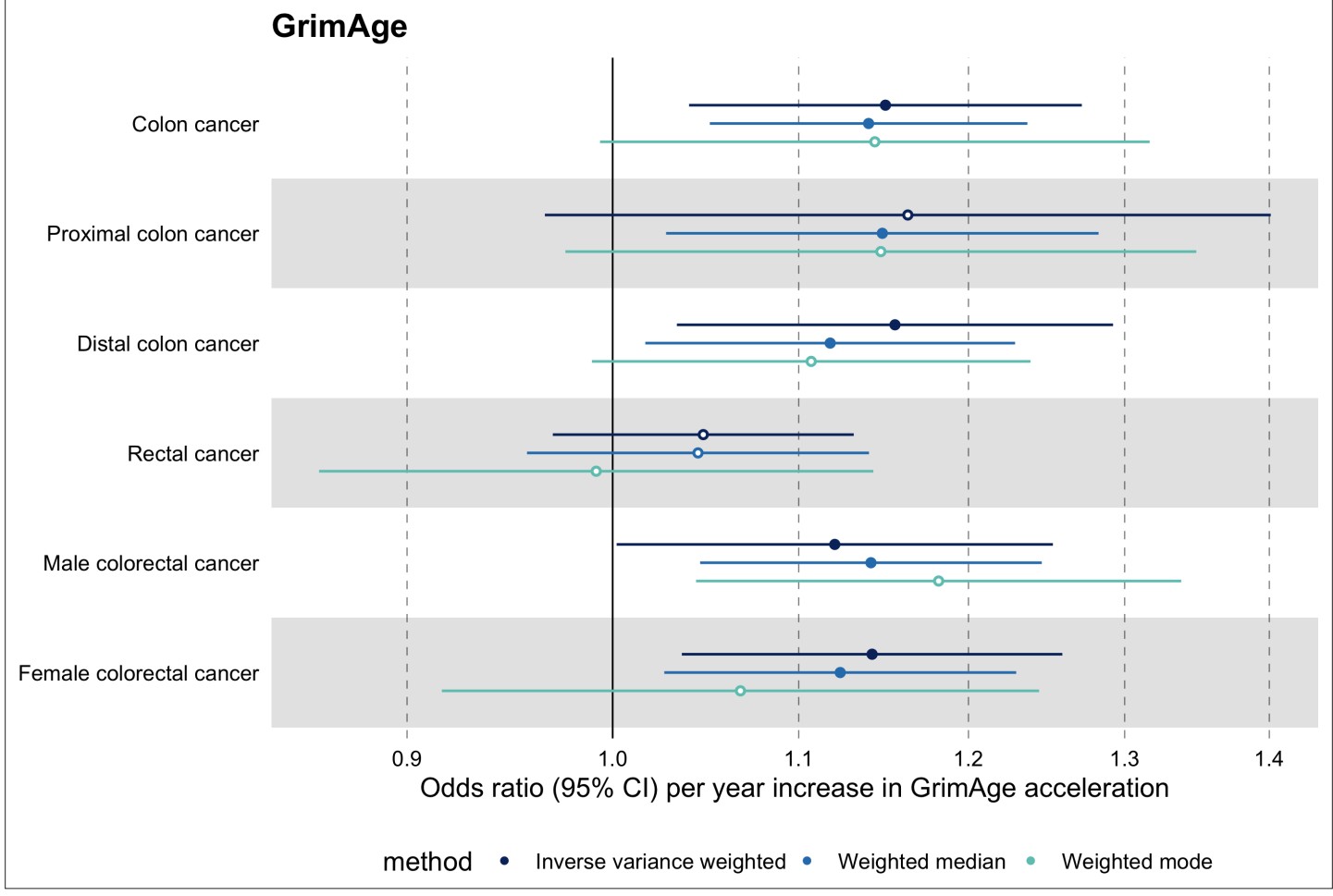

**Figure 5.** Mendelian randomization estimates for genetically predicted effects of GrimAge acceleration on colorectal cancer subtypes. Odds ratios and 95% confidence intervals are reported per 1 year increase in GrimAge acceleration. GrimAge acceleration was instrumented by four genetic variants. Results were obtained using inverse variance weighted MR (dark blue), weighted median (sky blue) and weighted mode (turquoise) methods. Data source: GECCO.

same for distal (IVW OR = 1.16, 95% CI 1.03–1.29, p = 0.01) and proximal colon cancer (IVW OR = 1.16, 95% CI 0.97–1.40, p = 0.11). Also, sex-stratified results suggest that GrimAge acceleration may influence colorectal cancer in both males (IVW OR = 1.12, 95% CI 1.00–1.25, p = 0.05) and females (IVW OR = 1.14, 95% CI 1.04–1.26, p = 0.008) (*Figure 5*, *Supplementary file 1 —*Table s11).

These findings were further supported by evidence of a positive association between GrimAge acceleration and parental history of colorectal cancer (OR = 1.06, 95% CI 1.00–1.12, p = 0.03) (*Figure 6*, *Supplementary file 1* — Table s12). Additionally, LD Score regression coefficients for GrimAge acceleration and colorectal cancer were also in the expected direction (GECCO rg = 0.28, p < 0.001; UK Biobank rg = 0.15, p = 0.21; FinnGen rg = 0.27, p = 0.29) (*Appendix 2—figure 12*, *Supplementary file 1* — Table s13).

We did not find consistent evidence of causality between other measures of epigenetic age acceleration and colorectal cancer (PhenoAge IVW OR = 1.00, 95% CI 0.97–1.02, p = 0.73; Hannum-Age IVW OR = 1.02, 95% CI 0.97–1.08, p = 0.37; and Intrinsic HorvathAge IVW OR = 1.00, 95% CI 0.97–1.02, p = 0.79) (*Figure 2*, *Appendix 2—figure 1*, *Appendix 2—figure 2*, *Appendix 2—figure 3*, *Appendix 2—figure 4*, *Appendix 2—figure 5*, *Appendix 2—figure 6*, *Appendix 2—figure 7*, *Appendix 2—figure 8*, *Supplementary file 1* — Table s3, *Supplementary file 1* — Table s4, *Supplementary file 1* — Table s5, *Supplementary file 1* — Table s6).

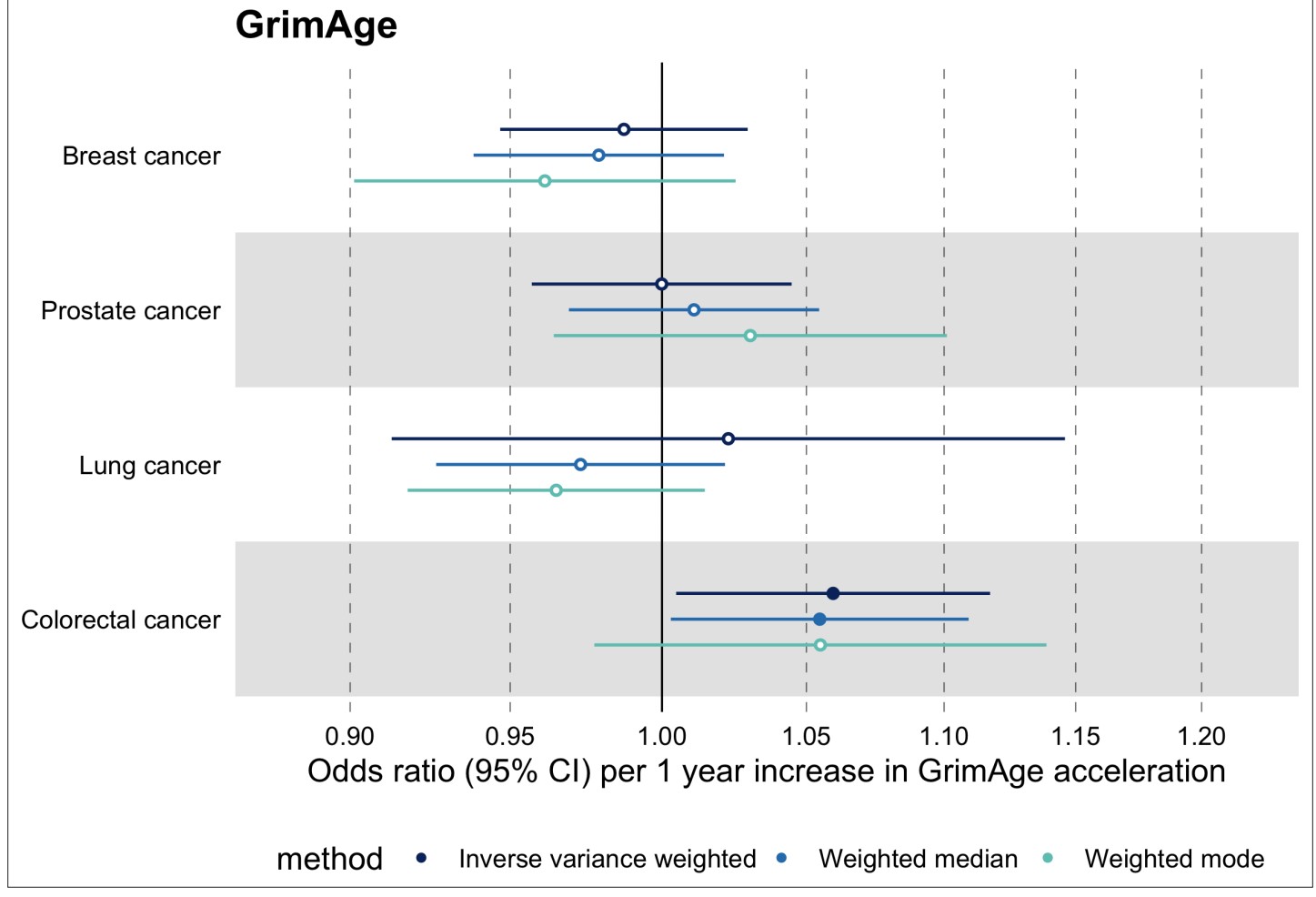

**Figure 6.** Mendelian randomization estimates for genetically predicted effects of GrimAge acceleration on parental history of multiple cancers. Odds ratios and 95% confidence intervals are reported per 1 year increase in GrimAge acceleration. GrimAge acceleration was instrumented by four genetic variants. Results were obtained using inverse variance weighted MR (dark blue), weighted median (sky blue) and weighted mode (turquoise) methods. Data source: UK Biobank.

### Telomere length

In bidirectional MR analyses, we found evidence that genetically predicted GrimAge acceleration may be a cause of telomere shortening (IVW beta coefficient = −0.07 per year increase in GrimAge acceleration, 95% CI –0.09 to –0.05, p < 0.001) and that genetically predicted longer telomere length may increase Intrinsic HorvathAge acceleration (IVW beta coefficient = 0.57 per standard deviation increase in telomere length, 95% CI 0.39–0.77, p = 0.002) (*Appendix 2—figure 13*, *Supplementary file 1* — Table s14).

Steiger filtering showed that all genetic instruments for GrimAge acceleration were stronger predictors of GrimAge acceleration than telomere length. In contrast, it identified 20 genetic instruments for telomere length that were better predictors of Intrinsic HorvathAge acceleration than telomere length (*Supplementary file 1* — Table s15). After removing these SNPs from the analyses, the results were still suggestive of an effect of telomere length on Intrinsic HorvathAge acceleration (IVW beta coefficient = 0.71 per standard deviation increase in telomere length, 95% CI 0.57–0.85, p < 0.001) (*Appendix 2—figure 14*, *Supplementary file 1* — Table s16).

There was little evidence of causality between other measures of epigenetic age acceleration and telomere length (*Appendix 2—figure 13*, *Supplementary file 1* — Table s14).

## Discussion

In this comprehensive two-sample MR study of epigenetic age acceleration and multiple cancers, we found evidence to suggest that genetically predicted GrimAge acceleration may increase the risk of colorectal cancer in both males and females. Among subtypes, effects appeared to be stronger in relation to colon than rectal cancer. Our MR results also suggested that genetically predicted GrimAge acceleration may decrease the risk of prostate cancer and that genetically predicted Intrinsic HorvathAge acceleration may be protective against lung cancer. Nevertheless, these did not pass multiple testing correction. Finally, we found no consistent evidence for other measures of epigenetic age acceleration and cancers.

Our MR estimates for the association between GrimAge and colorectal cancer were consistent with those reported in *Dugue et al., 2021*, an observational nested case-control study in the Melbourne Collaborative Cohort Study (RR = 1.04 per year increase in GrimAge acceleration, 95% CI 1.01–1.07, p = 0.02). However, our findings contrast with those highlighted in *Hillary et al., 2020*, an observational cohort study that used Generation Scotland data. The latter authors observed no evidence of an association between GrimAge acceleration and colorectal cancer after correcting for multiple testing. Nevertheless, it is possible that their analyses were underpowered, as their sample only included 63 colorectal cancer cases (0.66%). More importantly, the direction of the reported estimate is consistent with our findings and those presented in *Dugue et al., 2021*.

Observational evidence for the association between other measures of epigenetic ageing and cancer is inconclusive (the pre-existing evidence has been summarised in *Supplementary file 1* — Table s17). For instance, epigenetic clock acceleration has been positively associated with breast (*Ambatipudi et al., 2017*; *Kresovich et al., 2019b*; *Kresovich et al., 2019a*) and lung cancer (*Levine et al., 2018*; *Levine et al., 2015*; *Dugue et al., 2021*) in some studies. However, (*Durso et al., 2017*, *Hillary et al., 2020*) and (*Dugué et al., 2018*) did not find strong evidence to support this. In some cases, observational evidence is stronger for some clocks than it is for others. For example, for colorectal cancer, evidence of a positive association is much stronger for second-generation clocks (*Dugue et al., 2021*) than for first-generation clocks (*Dugué et al., 2018*; *Durso et al., 2017*). In the case of prostate cancer, as in our study, apart from weak evidence of an inverse association with GrimAge, no other associations have been observed (*Dugue et al., 2021*; *Dugué et al., 2018*). To date, the association between epigenetic age acceleration and ovarian cancer has not been explored observationally. Although our findings were less susceptible to biases that often influence observational research, they still provide no compelling evidence of a causality between several measures of epigenetic clock acceleration and cancer.

This MR study had several strengths. For instance, we pooled results from multiple sources using fixed effect meta-analysis to improve the precision of the MR estimates presented in *McCartney et al., 2021*. We also conducted extra sensitivity analyses, such as MR of negative control outcomes, MR of cancer risk factors, single-SNP MR and CAUSE analyses, to assess the validity of the MR assumptions. Moreover, we performed subtype-specific MR analyses and sought to corroborate our results using UK Biobank GWAS data on parental history of cancer and LD Score regression. Additionally, our findings contribute to the identification of modifiable targets for future interventions aimed at reversing epigenetic ageing for the prevention of cancer. Compared to clinical trials, MR provides a cheaper, quicker, and ethical way of assessing the long-term impact of interventions on epigenetic ageing. This is especially relevant while attempts to develop interventions which reverse epigenetic ageing are still in early stages (*Fahy et al., 2019*; *Fitzgerald et al., 2021*; *Gensous et al., 2020*; *Chen et al., 2019*).

The findings from this study should be interpreted in light of its limitations. We only identified four genetic instruments for GrimAge acceleration, which explained 0.47% of the variance in the trait. This could lead to two issues: low statistical power and horizontal pleiotropy. First, our GrimAge analyses were underpowered to detect ORs < 1.20 for colorectal cancer. Therefore, it is possible that our findings do not reflect a true effect (we identified an OR = 1.12 for colorectal). Similarly, our study was underpowered to detect genetically predicted effects of GrimAge acceleration on cancer subtypes and cancers with smaller sample sizes (i.e. ovarian and lung cancer). Some of our sensitivity analyses, such as the MR-Egger intercept test used to detect uncorrelated horizontal pleiotropy, also had low power, resulting in imprecise estimates. The weighted mode method may also be misleading in this context, as its use is limited in the presence of very few SNPs. Although these limitations potentially undermine the validity of our results, it is reassuring that point estimates for the genetically

predicted effect of GrimAge acceleration on colorectal cancer were consistent across MR methods and study populations. However, since CAUSE analyses did not provide evidence against confounding by correlated horizontal pleiotropy, it is possible that the genetically predicted effects identified are attributed to correlated pleiotropy (whereby SNPs are associated with epigenetic age acceleration and cancer through a shared heritable factor) rather than a causal effect of GrimAge on cancer risk.

One could argue that because the results for GrimAge acceleration were inconsistent with those obtained for other measures of epigenetic age acceleration, chance and horizontal pleiotropy are more likely explanations for our findings. However, inconsistencies across epigenetic ageing measures do not necessarily invalidate our results. They may simply reflect differences in how clocks were trained (i.e. they were trained on different outcomes, tissues, and populations). Different clocks may capture information on distinct underlying biological ageing mechanisms (*Liu et al., 2020*). For example, GrimAge was trained on mortality and smoking (factors which are closely related to cancer risk), which may explain why it outperforms other measures of epigenetic ageing in predicting time-to-cancer (*Lu et al., 2019a*).

Although little is known about the underlying mechanisms, GrimAge may plausibly influence cancer risk through hormonal, inflammatory and metabolic processes (*Yu et al., 2020*; *Bottazzi et al., 2018*; *Lau and Robinson, 2021*). In bidirectional MR analyses, we found evidence that genetically predicted GrimAge acceleration may be a cause of telomere shortening, another marker of biological ageing. Shorter telomeres have been shown to lower cancer risk in prior MR analyses (*Telomeres Mendelian Randomization Collaboration et al., 2017*; *Gao et al., 2020*; *Kuo et al., 2019*), so it is plausible that GrimAge acceleration decreases cancer risk, at least in part, via its effect on telomere length. GrimAge acceleration may still increase cancer risk via pathways other than those related to cellular division. The positive effect of GrimAge acceleration on cancer via these other pathways may counteract the negative effects mediated via telomere length, resulting in null MR results for GrimAge acceleration and breast, ovarian, prostate and lung cancer, and positive MR results for GrimAge acceleration and colorectal cancer. To better understand the biology of ageing, future studies should consider running MR analyses using data on DNAm-predicted telomere length, since DNAm telomere length is independent of telomerase activity and has been more strongly associated with age than measured telomere length (*Lu et al., 2019b*).

Although promising in terms of consistency and biological plausibility, further research is required to confirm our findings. For example, multivariable MR (*Burgess and Thompson, 2015*; *Sanderson et al., 2019*) could be used to disentangle the causal effects of GrimAge acceleration on cancer from shared heritable factors such as and blood cell composition. Additionally, our analyses could be replicated using other large independent cancer datasets. Although we conducted MR analyses on parental history of cancer, their effect estimates are not directly comparable to those obtained in the main analyses due to cases in the GWAS-by-proxy of parental endpoints being defined as either or both parents reportedly having a type of cancer. Furthermore, it would also be useful to replicate our analyses once a larger GWAS of epigenetic ageing with more genetic instruments for GrimAge acceleration is available. This would allow for a more rigorous assessment of horizontal pleiotropy and may be used to assess clustering of genetic variants to reveal distinct biological mechanisms underlying the effects (*Foley et al., 2021*). In spite of these suggestions, we acknowledge that it may be challenging to get access to suitable datasets for replication purposes in the short term.

The selection of 'super controls' (e.g. in UK Biobank, FinnGen and GECCO), with no other cancers, related lesions (i.e. benign, in situ, uncertain or unspecified behaviour neoplasms) or reported family history of cancer, could have inflated cancer GWAS effect sizes (and our MR estimates), because 'super controls' are healthier than the general population and are less likely to be genetically predisposed to develop cancer.

Another limitation is that we did not have access to individual level data. Therefore, we were unable to stratify the analyses by potential effect modifiers, such as sex, smoking, and menopausal status. Moreover, we did not have sex-specific instruments for sex-specific cancers. However, it is unlikely that the genetic architecture of epigenetic clock acceleration differs across sexes, as DNAm levels at individual clock CpGs are highly correlated between males and females (*Grodstein et al., 2020*; *Tajuddin et al., 2019*).

Finally, to reduce bias due to population stratification, this study was conducted using data from participants of European ancestry only. The GWAS data used for the analyses had been adjusted for

the top genetic principal components for the same reason. Furthermore, our MR of negative control outcomes suggests that our MR results are unlikely to be biased by residual population stratification. Despite this, confounding due to population stratification, dynastic effects and assortative mating cannot be ruled out completely, as it is not possible to test the second MR assumption (i.e. independence assumption). Furthermore, more research is required to see if our results could translate to other ancestries.

From a public health perspective, our work provides potentially relevant findings. Observational and Mendelian randomization studies suggest that GrimAge acceleration may be influenced by several cancer risk factors, such as obesity and smoking (*Lu et al., 2019a*; *McCartney et al., 2021*). If GrimAge acceleration is a causal mediator between these risk factors and colorectal cancer, the GrimAge clock may be a treatable intermediary when targeting the underlying risk factors is not feasible or too difficult to accomplish. It could also be targeted in populations at high-risk of colorectal cancer. Nevertheless, we think it may be too early to make claims regarding the clinical utility of our findings. The GrimAge clock has only recently been created and very few studies have assessed its association with colorectal cancer. More research is required to corroborate our results and to evaluate whether GrimAge acceleration can be modified through lifestyle or clinical interventions.

In conclusion, our findings suggest that genetically predicted GrimAge acceleration may increase the risk of colorectal cancer. Findings were less consistent for other epigenetic clocks and cancers. Further work is required to investigate the potential mechanisms underlying the genetically predicted effects identified in this study.

## Acknowledgements

We thank Richard Wilkinson for proofreading several versions of the manuscript and Dr Jean Morrison for helping us interpret the CAUSE analysis output. We acknowledge the participants and investigators of the FinnGen and UK Biobank studies. GWAS data on parental history of cancer were generated using the UK Biobank Resource under application number 15825. Finally, we thank the BCAC, OCAC, PRACTICAL, ILCCO, GECCO and CIMBA consortiums for their contributions.

CRUK and PRACTICAL consortium Funding Acknowledgements: This work was supported by the Canadian Institutes of Health Research, European Commission's Seventh Framework Programme grant agreement n° 223175 (HEALTH-F2-2009-223175), Cancer Research UK Grants C5047/A7357, C1287/A10118, C1287/A16563, C5047/A3354, C5047/A10692, C16913/A6135, and The National Institute of Health (NIH) Cancer Post-Cancer GWAS initiative grant: No. 1 U19 CA 148537-01 (the GAME-ON initiative).

We also thank the following for funding support: The Institute of Cancer Research and The Everyman Campaign, The Prostate Cancer Research Foundation, Prostate Research Campaign UK (now PCUK), The Orchid Cancer Appeal, Rosetrees Trust, The National Cancer Research Network UK, The National Cancer Research Institute (NCRI) UK. We are grateful for support of NIHR funding to the NIHR Biomedical Research Centre at The Institute of Cancer Research, The Royal Marsden NHS Foundation Trust, and Manchester NIHR Biomedical Research Centre. The Prostate Cancer Program of Cancer Council Victoria also acknowledge grant support from The National Health and Medical Research Council, Australia (126402, 209057, 251533, , 396414, 450104, 504700, 504702, 504715, 623204, 940394, 614296,), VicHealth, Cancer Council Victoria, The Prostate Cancer Foundation of Australia, The Whitten Foundation, PricewaterhouseCoopers, and Tattersall's. EAO, DMK, and EMK acknowledge the Intramural Program of the National Human Genome Research Institute for their support.

Genotyping of the OncoArray was funded by the US National Institutes of Health (NIH) [U19 CA 148537 for ELucidating Loci Involved in Prostate cancer SuscEptibility (ELLIPSE) project and X01HG007492 to the Center for Inherited Disease Research (CIDR) under contract number HHSN268201200008I]. Additional analytic support was provided by NIH NCI U01 CA188392 (PI: Schumacher).

Research reported in this publication also received support from the National Cancer Institute of the National Institutes of Health under Award Numbers U10 CA37429 (CD Blanke), and UM1 CA182883 (CM Tangen/IM Thompson). The content is solely the responsibility of the authors and does not necessarily represent the official views of the National Institutes of Health.

Funding for the iCOGS infrastructure came from: the European Community's Seventh Framework Programme under grant agreement n° 223175 (HEALTH-F2-2009-223175) (COGS), Cancer Research UK (C1287/A10118, C1287/A 10710, C12292/A11174, C1281/A12014, C5047/A8384, C5047/A15007, C5047/A10692, C8197/A16565), the National Institutes of Health (CA128978) and Post-Cancer GWAS initiative (1U19 CA148537, 1U19 CA148065 and 1U19 CA148112 - the GAME-ON initiative), the Department of Defence (W81XWH-10-1-0341), the Canadian Institutes of Health Research (CIHR) for the CIHR Team in Familial Risks of Breast Cancer, Komen Foundation for the Cure, the Breast Cancer Research Foundation, and the Ovarian Cancer Research Fund.

## Additional information

### Group author details

the PRACTICAL consortium

**Rosalind A Eeles**: The Institute of Cancer Research, London, United Kingdom; Royal Marsden NHS Foundation Trust, London, United Kingdom; **Christopher A Haiman**: Center for Genetic Epidemiology, Department of Preventive Medicine, Keck School of Medicine, University of Southern California/Norris Comprehensive Cancer Center, Los Angeles, United States; **Zsofia Kote-Jarai**: The Institute of Cancer Research, London, United Kingdom; **Fredrick R Schumacher**: Department of Population and Quantitative Health Sciences, Case Western Reserve University, Cleveland, United States; Seidman Cancer Center, University Hospitals, Cleveland, United States; **Sara Benlloch**: The Institute of Cancer Research, London, United Kingdom; Centre for Cancer Genetic Epidemiology, Department of Public Health and Primary Care, University of Cambridge, Strangeways Research Laboratory, Cambridge, United Kingdom; **Ali Amin Al Olama**: Centre for Cancer Genetic Epidemiology, Department of Public Health and Primary Care, University of Cambridge, Strangeways Research Laboratory, Cambridge, United Kingdom; University of Cambridge, Department of Clinical Neurosciences, Stroke Research Group, Cambridge, United Kingdom; **Kenneth R Muir**: Division of Population Health, Health Services Research and Primary Care, University of Manchester, Manchester, United Kingdom; **Sonja I Berndt**: Division of Cancer Epidemiology and Genetics, National Cancer Institute, NIH, Bethesda, United States; **David V Conti**: Center for Genetic Epidemiology, Department of Preventive Medicine, Keck School of Medicine, University of Southern California/Norris Comprehensive Cancer Center, Los Angeles, United States; **Fredrik Wiklund**: Department of Medical Epidemiology and Biostatistics, Karolinska Institute, Stockholm, Sweden; **Stephen Chanock**: Division of Cancer Epidemiology and Genetics, National Cancer Institute, NIH, Bethesda, United States; **Ying Wang**: Department of Population Science, American Cancer Society, Atlanta, United States; **Catherine M Tangen**: SWOG Statistical Center, Fred Hutchinson Cancer Research Center, Seattle, United States; **Jyotsna Batra**: Australian Prostate Cancer Research Centre-Qld, Institute of Health and Biomedical Innovation and School of Biomedical Sciences, Queensland University of Technology, Brisbane, Australia; Translational Research Institute, Brisbane, Australia; **Judith A Clements**: Australian Prostate Cancer Research Centre-Qld, Institute of Health and Biomedical Innovation and School of Biomedical Sciences, Queensland University of Technology, Brisbane, Australia; Translational Research Institute, Brisbane, Australia; **APCB BioResource (Australian Prostate Cancer BioResource)**: Translational Research Institute, Brisbane, Australia; Australian Prostate Cancer Research Centre-Qld, Queensland University of Technology, Brisbane; Prostate Cancer Research Program, Monash University, Melbourne; Dame Roma Mitchell Cancer Centre, University of Adelaide, Adelaide, Australia; **Henrik Grönberg**: Department of Medical Epidemiology and Biostatistics, Karolinska Institute, Stockholm, Sweden; **Nora Pashayan**: Department of Applied Health Research, University College London, London, United Kingdom; Centre for Cancer Genetic Epidemiology, Department of Oncology, University of Cambridge, Strangeways Laboratory, Worts Causeway, Cambridge, United Kingdom; **Johanna Schleutker**: Institute of Biomedicine, University of Turku, Turku, Finland; Department of Medical Genetics, Genomics, Laboratory Division, Turku University Hospital, Turku, Finland; **Demetrius Albanes**: Division of Cancer Epidemiology and Genetics, National Cancer Institute, NIH, Bethesda, United States; **Stephanie Weinstein**: Division of Cancer Epidemiology and Genetics, National Cancer Institute, NIH, Bethesda,

United States; **Alicja Wolk**: Department of Surgical Sciences, Uppsala University, Uppsala, Sweden; **Catharine ML West**: Division of Cancer Sciences, University of Manchester, Manchester Academic Health Science Centre, Radiotherapy Related Research, The Christie Hospital NHS Foundation Trust, Manchester, United Kingdom; **Lorelei A Mucci**: Department of Epidemiology, Harvard T. H. Chan School of Public Health, Boston, United States; **Géraldine Cancel-Tassin**: CeRePP, Tenon Hospital, Paris, France; Sorbonne Universite, GRC n°5 , AP-HP, Tenon Hospital, 4 rue de la Chine, Paris, France; **Stella Koutros**: Division of Cancer Epidemiology and Genetics, National Cancer Institute, NIH, Bethesda, United States; **Karina Dalsgaard Sørensen**: Department of Molecular Medicine, Aarhus University Hospital, Aarhus, Denmark; Department of Clinical Medicine, Aarhus University, Aarhus, Denmark; **Eli Marie Grindedal**: Department of Medical Genetics, Oslo University Hospital, Oslo, Norway; **David E Neal**: Nuffield Department of Surgical Sciences, University of Oxford, Room 6603, Level 6, John Radcliffe Hospital, Headley Way, Oxford, United Kingdom; University of Cambridge, Department of Oncology, Cambridge, United Kingdom; Cancer Research UK, Cambridge Research Institute, Cambridge, United Kingdom; **Freddie C Hamdy**: Nuffield Department of Surgical Sciences, University of Oxford, Oxford, United Kingdom; Faculty of Medical Science, University of Oxford, John Radcliffe Hospital, Oxford, United Kingdom; **Jenny L Donovan**: Population Health Sciences, Bristol Medical School, University of Bristol, Bristol, United Kingdom; **Ruth C Travis**: Cancer Epidemiology Unit, Nuffield Department of Population Health, University of Oxford, Oxford, United Kingdom; **Robert J Hamilton**: Dept. of Surgical Oncology, Princess Margaret Cancer Centre, Toronto, Canada; Dept. of Surgery (Urology), University of Toronto, Toronto, Canada; **Sue Ann Ingles**: Department of Preventive Medicine, Keck School of Medicine, University of Southern California/Norris Comprehensive Cancer Center, Los Angeles, United States; **Barry S Rosenstein**: Department of Radiation Oncology and Department of Genetics and Genomic Sciences, Box 1236, Icahn School of Medicine at Mount Sinai, One Gustave L. Levy Place, New York, United States; **Yong-Jie Lu**: Centre for Cancer Biomarker and Biotherapeutics, Barts Cancer Institute, Queen Mary University of London, John Vane Science Centre, Charterhouse Square, London, United Kingdom; **Graham G Giles**: Cancer Epidemiology Division, Cancer Council Victoria, Melbourne, Australia; Centre for Epidemiology and Biostatistics, Melbourne School of Population and Global Health, The University of Melbourne, Parkville, Australia; Precision Medicine, School of Clinical Sciences at Monash Health, Monash University, Victoria, Australia; **Robert J MacInnis**: Cancer Epidemiology Division, Cancer Council Victoria, Melbourne, Australia; Centre for Epidemiology and Biostatistics, Melbourne School of Population and Global Health, The University of Melbourne, Parkville, Australia; **Adam S Kibel**: Division of Urologic Surgery, Brigham and Womens Hospital, Boston, United States; **Ana Vega**: Fundación Pública Galega Medicina Xenómica, Santiago de Compostela, Spain; Instituto de Investigación Sanitaria de Santiago de Compostela, Santiago De Compostela, Spain; Centro de Investigación en Red de Enfermedades Raras (CIBERER), Valencia, Spain; **Manolis Kogevinas**: ISGlobal, Barcelona, Spain; IMIM (Hospital del Mar Medical Research Institute), Barcelona, Spain; Universitat Pompeu Fabra (UPF), Barcelona, Spain; CIBER Epidemiología y Salud Pública (CIBERESP), Madrid, Spain; **Kathryn L Penney**: Channing Division of Network Medicine, Department of Medicine, Brigham and Women's Hospital/Harvard Medical School, Boston, United States; **Jong Y Park**: Department of Cancer Epidemiology, Moffitt Cancer Center, Tampa, United States; **Janet L Stanford**: Division of Public Health Sciences, Fred Hutchinson Cancer Research Center, Seattle, United States; Department of Epidemiology, School of Public Health, University of Washington, Seattle, United States; **Cezary Cybulski**: International Hereditary Cancer Center, Department of Genetics and Pathology, Pomeranian Medical University, Szczecin, Poland; **Børge G Nordestgaard**: Faculty of Health and Medical Sciences, University of Copenhagen, Copenhagen, Denmark; Department of Clinical Biochemistry, Herlev and Gentofte Hospital, Copenhagen University Hospital, Herlev, Denmark; **Sune F Nielsen**: Faculty of Health and Medical Sciences, University of Copenhagen, Copenhagen, Denmark; Department of Clinical Biochemistry, Herlev and Gentofte Hospital, Copenhagen University Hospital, Herlev, Denmark; **Hermann Brenner**: Division of Clinical Epidemiology and Aging Research, German Cancer Research Center (DKFZ), Heidelberg, Germany; German Cancer Consortium (DKTK), German Cancer Research Center (DKFZ), Heidelberg, Germany; Division of Preventive Oncology, German Cancer Research Center (DKFZ) and National Center for Tumor Diseases (NCT), Heidelberg, Germany; **Christiane Maier**: Humangenetik Tuebingen,

Tuebingen, Germany; **UJeri Kim**: The University of Texas M. D. Anderson Cancer Center, Department of Genitourinary Medical Oncology, Houston, United States; **Esther M John**: Departments of Epidemiology & Population Health and of Medicine, Division of Oncology, Stanford Cancer Institute, Stanford University School of Medicine, Stanford, United States; **Manuel R Teixeira**: Department of Genetics, Portuguese Oncology Institute of Porto (IPO-Porto), Porto, Portugal; Biomedical Sciences Institute (ICBAS), University of Porto, Porto, Portugal; Cancer Genetics Group, IPO-Porto Research Center (CI-IPOP), Portuguese Oncology Institute of Porto (IPO-Porto), Porto, Portugal; **Susan L Neuhausen**: Department of Population Sciences, Beckman Research Institute of the City of Hope, Duarte, United States; **Kim De Ruyck**: Ghent University, Faculty of Medicine and Health Sciences, Basic Medical Sciences, Ghent, Belgium; **Azad Razack**: Department of Surgery, Faculty of Medicine, University of Malaya, Kuala Lumpur, Malaysia; **Lisa F Newcomb**: Division of Public Health Sciences, Fred Hutchinson Cancer Research Center, Seattle, United States; Department of Urology, University of Washington, Seattle, United States; **Davor Lessel**: Institute of Human Genetics, University Medical Center Hamburg-Eppendorf, Hamburg, Germany; **Radka Kaneva**: Molecular Medicine Center, Department of Medical Chemistry and Biochemistry, Medical University of Sofia, Sofia, Bulgaria; **Nawaid Usmani**: Department of Oncology, Cross Cancer Institute, University of Alberta, Edmonton, Canada; Division of Radiation Oncology, Cross Cancer Institute, University Avenue, Edmonton, Canada; **Frank Claessens**: Molecular Endocrinology Laboratory, Department of Cellular and Molecular Medicine, Leuven, Belgium; **Paul A Townsend**: Division of Cancer Sciences, Manchester Cancer Research Centre, Faculty of Biology, Medicine and Health, Manchester Academic Health Science Centre, NIHR Manchester Biomedical Research Centre, Health Innovation Manchester, Univeristy of Manchester, Manchester, United Kingdom; The University of Surrey, Guildford, Surrey, United Kingdom; **Jose Esteban Castelao**: Genetic Oncology Unit, CHUVI Hospital, Complexo Hospitalario Universitario de Vigo, Instituto de Investigación Biomédica Galicia Sur (IISGS), Vigo, Spain; **Monique J Roobol**: Department of Urology, Erasmus University Medical Center, Rotterdam, Netherlands; **Florence Menegaux**: "Exposome and Heredity", CESP (UMR 1018), Faculté de Médecine, Université Paris-Saclay, Villejuif, France; **Kay-Tee Khaw**: Clinical Gerontology Unit, University of Cambridge, Cambridge, United Kingdom; **ULisa Cannon-Albright**: Division of Epidemiology, Department of Internal Medicine, University of Utah School of Medicine, Salt Lake City, United States; George E. Wahlen Department of Veterans Affairs Medical Center, Salt Lake City, United States; **Hardev Pandha**: The University of Surrey, Guildford, Surrey, United Kingdom; **Stephen N Thibodeau**: Department of Laboratory Medicine and Pathology, Mayo Clinic, Rochester, United States; **David J Hunter**: Nuffield Department of Population Health, University of Oxford, Oxford, United Kingdom; **Peter Kraft**: Program in Genetic Epidemiology and Statistical Genetics, Department of Epidemiology, Harvard School of Public Health, Boston, United States; **William J Blot**: Division of Epidemiology, Department of Medicine, Vanderbilt University Medical Center, Nashville, United States; International Epidemiology Institute, Rockville, United States; **Elio Riboli**: Department of Epidemiology and Biostatistics, School of Public Health, Imperial College London, London, United Kingdom

## Competing interests

Ake T Lu: declares that UC Regents filed the patent "DNA METHYLATION BASED BIOMARKERS FOR LIFE EXPECTANCY AND MORBIDITY" (International Application Number PCT/US2019/055444; in pending status) and that the Epigenetic Clock Development Foundation and Foxo Labs hold licenses. the PRACTICAL consortium: Steve Horvath: declares that UC Regents filed the patent "DNA METHYLATION BASED BIOMARKERS FOR LIFE EXPECTANCY AND MORBIDITY" (International Application Number PCT/US2019/055444; in pending status) and that the Epigenetic Clock Development Foundation and Foxo Labs hold licenses. SH receives consulting fees from the Epigenetic Clock Development Foundation and royalties for patents involving epigenetic clocks. Riccardo E Marioni: has received a speaker fee from Illumina and is an advisor to the Epigenetic Clock Development Foundation. Tom G Richardson: is employed part time by Novo Nordisk outside of this work. The other authors declare that no competing interests exist.

## Funding

| Funder | Grant reference number | Author |
|---|---|---|
| Wellcome Trust | 224982/Z/22/Z | Fernanda Morales Berstein |
| Cancer Research UK | C18281/A29019 | Konstantinos K Tsilidis<br>Philip C Haycock<br>Richard M Martin<br>Caroline L Relton<br>George Davey Smith |
| Hellenic Republic's Operational Programme "Competitiveness, Entrepreneurship & Innovation" | ΟΠΣ 5047228 | Konstantinos K Tsilidis |
| NIHR Biomedical Research Centre at University Hospitals Bristol | | Richard M Martin |
| Weston NHS Foundation Trust | | Richard M Martin |
| NIHR Senior Investigator | NIHR202411 | Richard M Martin |
| Medical Research Council | MC_UU_00011/5 and MC_UU_00011/1 | Caroline L Relton<br>George Davey Smith |
| Alzheimer's Society | AS-PG-19b-010 | Riccardo E Marioni |
| National Institutes of Health | U01 AG-18-018 | Riccardo E Marioni<br>Steve Horvath |
| de Pass Vice Chancellor's Research Fellow at the University of Bristol | | Rebecca C Richmond |

The funders had no role in study design, data collection and interpretation, or the decision to submit the work for publication.

## Author contributions

Fernanda Morales Berstein, Conceptualization, Formal analysis, Visualization, Writing – original draft, Writing – review and editing; Daniel L McCartney, Ake T Lu, Konstantinos K Tsilidis, Philip Haycock, Kimberley Burrows, Amanda I Phipps, Daniel D Buchanan, Iona Cheng, Richard M Martin, George Davey Smith, Caroline L Relton, Steve Horvath, Riccardo E Marioni, Writing – review and editing; Emmanouil Bouras, Formal analysis, Writing – review and editing; the PRACTICAL consortium, Contributions to the acquisition of data, critical revision of the manuscript and final approval of the version to be published; Tom G Richardson, Conceptualization, Formal analysis, Supervision, Writing – review and editing; Rebecca C Richmond, Conceptualization, Supervision, Writing – review and editing

## Author ORCIDs

Fernanda Morales Berstein ⓘ http://orcid.org/0000-0002-8237-2021
Philip Haycock ⓘ http://orcid.org/0000-0001-5001-3350
Richard M Martin ⓘ http://orcid.org/0000-0002-7992-7719
George Davey Smith ⓘ http://orcid.org/0000-0002-1407-8314
Caroline L Relton ⓘ http://orcid.org/0000-0003-2052-4840
Riccardo E Marioni ⓘ http://orcid.org/0000-0003-4430-4260
Tom G Richardson ⓘ http://orcid.org/0000-0002-7918-2040

## Ethics

Human subjects: This research did not require ethical approval as it used secondary, genome-wide association data from studies that obtained informed consent from all participants and ethical approval from review boards and/or ethics committees.

## Decision letter and Author response

Decision letter https://doi.org/10.7554/eLife.75374.sa1
Author response https://doi.org/10.7554/eLife.75374.sa2

## Additional files

**Supplementary files**

• Supplementary file 1. Supplementary Tables 1-17. Table s1. Genetic instruments for epigenetic age acceleration. Table s2. Power calculations. Table s3. MR for genetically predicted epigenetic age acceleration and cancer (international consortia). Table s4. MR for genetically predicted epigenetic age acceleration and cancer (UK Biobank). Table s5. MR for genetically predicted epigenetic age acceleration and cancer (FinnGen). Table s6. Fixed effect meta-analysis of MR for genetically predicted epigenetic age acceleration and cancer (Consortia, UK Biobank and FinnGen). Table s7. Mendelian randomization analyses for the genetically predicted effects of epigenetic clock acceleration on negative control outcomes. Table s8. Mendelian randomization analyses for the genetically predicted effects of epigenetic clock acceleration on potential confounders of the association between epigenetic age acceleration and cancer. Table s9. Single SNP analyses for the genetically predicted effects of epigenetic clock acceleration on multiple cancers. Table s10. Cochran's Q and MR-Egger Intercept for the genetically predicted effects of epigenetic age acceleration on multiple cancers. Table s11. MR for genetically predicted epigenetic age acceleration and cancer subtypes (international consortia). Table s12. MR for genetically predicted epigenetic age acceleration and parental history of cancer (UK Biobank). Table s13. LD Score regression results for genetically predicted epigenetic age acceleration and cancer. Table s14. Bidirectional MR of epigenetic age acceleration and telomere length. Table s15. Steiger filtering for Bidirectional MR of epigenetic age acceleration and telomere length. Table s16. Bidirectional MR of epigenetic age acceleration and telomere length after Steiger filtering. Table s17. Preexisting evidence on epigenetic age acceleration and cancer.

• Supplementary file 2. STROBE-MR checklist of recommended items to address in reports of *Mendelian randomization studies*.

• Transparent reporting form

**Data availability**

Summary statistics for epigenetic age acceleration measures of HannumAge, Intrinsic HorvathAge, PhenoAge and GrimAge were downloaded from: https://datashare.ed.ac.uk/handle/10283/3645 (Datasets used: European-ancestries meta-analysis summary statistics: Hannum (645.4Mb), European-ancestries meta-analysis summary statistics: IEAA (645.7Mb), European-ancestries meta-analysis summary statistics: GrimAge (645.7Mb), European-ancestries meta-analysis summary statistics: PhenoAge (645.7Mb)). Summary statistics for international cancer genetic consortiums were obtained from their respective data repositories. Colorectal cancer data were obtained following the submission of a written request to the GECCO committee, which may be contacted by email at kafdem@fredhutch.org/upeters@fredhutch.org. Breast, ovarian, prostate and lung cancer data were accessed via MR-Base (http://app.mrbase.org/), which holds complete GWAS summary data from BCAC, OCAC, PRACTICAL and ILCCO. Breast cancer subtype data were obtained from BCAC and can be downloaded from: http://bcac.ccge.medschl.cam.ac.uk/bcacdata/oncoarray/oncoarray-and-combined-summary-result/gwas-summary-associations-breast-cancer-risk-2020/. Data on breast and ovarian cancer in BRCA1 and BRCA2 carriers were obtained from CIMBA and can be downloaded from: http://cimba.ccge.medschl.cam.ac.uk/oncoarray-complete-summary-results/. Prostate cancer subtype data are not publicly available through MR-Base but can be accessed upon request. These data are managed by the PRACTICAL committee, which may be contacted by email at practical@icr.ac.uk. FinnGen data is publicly available and can be accessed here: https://www.finngen.fi/en/access_results (Datasets used from release 5: Malignant neoplasm of breast (all cancers excluded), Malignant neoplasm of bronchus and lung (all cancers excluded), Colorectal cancer (all cancers excluded), Malignant neoplasm of ovary (all cancers excluded), Malignant neoplasm of prostate (all cancers excluded)). UK Biobank data can be accessed through the MR-Base platform. Parental history of cancer data were obtained from the UK Biobank study under application #15825 and can be accessed via an approved application to the UK Biobank (https://www.ukbiobank.ac.uk/enable-your-research/apply-for-access). GWAS data for negative control outcomes and potential confounders were obtained via the MR-Base platform (GWAS IDs for negative control outcomes: "ukb-b-19560", "ukb-b-533"; GWAS IDs for confounders: "ukb-b-10831", "ukb-b-13702", "ukb-b-6134", "ieu-a-1239", "ukb-b-5779", "ieu-a-835", "ieu-a-61"). GWAS data for measured telomere length used in bidirectional MR analyses were also obtained via the MR-Base platform (GWAS ID: "ieu-b-4879").

The following previously published datasets were used:

| Author(s) | Year | Dataset title | Dataset URL | Database and Identifier |
|---|---|---|---|---|
| McCartney DL, Min JL, Richmond RC | 2021 | Genome-wide association studies identify 137 loci for DNA methylation biomarkers of aging | https://doi.org/10.7488/ds/2834 | Edinburgh DataShare, 10.7488/ds/2834 |
| Phelan, et al | 2017 | Ovarian cancer | https://gwas.mrcieu.ac.uk/datasets/ieu-a-1120/ | IEU OpenGWAS, ieu-a-1120 |
| Schumacher, et al | 2018 | Prostate cancer | https://gwas.mrcieu.ac.uk/datasets/ieu-b-85/ | IEU OpenGWAS, ieu-b-85 |
| Michailidou K | 2017 | Breast cancer (Combined Oncoarray; iCOGS; GWAS meta analysis) | https://gwas.mrcieu.ac.uk/datasets/ieu-a-1126/ | IEU OpenGWAS, ieu-a-1126 |
| Wang Y | 2014 | Lung cancer | https://gwas.mrcieu.ac.uk/datasets/ieu-a-966/ | IEU OpenGWAS, ieu-a-966 |
| Wang Y | 2014 | Lung adenocarcinoma | https://gwas.mrcieu.ac.uk/datasets/ieu-a-965/ | IEU OpenGWAS, ieu-a-965 |
| Wang Y | 2014 | Squamous cell lung cancer | https://gwas.mrcieu.ac.uk/datasets/ieu-a-967/ | IEU OpenGWAS, ieu-a-967 |
| Phelan, et al | 2017 | High grade serous ovarian cancer | https://gwas.mrcieu.ac.uk/datasets/ieu-a-1121/ | IEU OpenGWAS, ieu-a-1121 |
| Phelan, et al | 2017 | Low grade serous ovarian cancer | https://gwas.mrcieu.ac.uk/datasets/ieu-a-1122/ | IEU OpenGWAS, ieu-a-1122 |
| Phelan, et al | 2017 | Invasive mucinous ovarian cancer | https://gwas.mrcieu.ac.uk/datasets/ieu-a-1123/ | IEU OpenGWAS, ieu-a-1123 |
| Phelan, et al | 2017 | Clear cell ovarian cancer | https://gwas.mrcieu.ac.uk/datasets/ieu-a-1124/ | IEU OpenGWAS, ieu-a-1124 |
| Phelan, et al | 2017 | Endometrioid ovarian cancer | https://gwas.mrcieu.ac.uk/datasets/ieu-a-1125/ | IEU OpenGWAS, ieu-a-1125 |
| Michailidou K | 2017 | ER+ Breast cancer (Combined Oncoarray; iCOGS; GWAS meta analysis) | https://gwas.mrcieu.ac.uk/datasets/ieu-a-1127/ | IEU OpenGWAS, ieu-a-1127 |
| Michailidou K | 2017 | ER- Breast cancer (Combined Oncoarray; iCOGS; GWAS meta analysis) | https://gwas.mrcieu.ac.uk/datasets/ieu-a-1128/ | IEU OpenGWAS, ieu-a-1128 |
| FinnGen consortium | 2021 | r5.finngen.fi | https://www.finngen.fi/en/access_results | FinnGen public data r5, r5.finngen.fi |
| Codd, et al | 2021 | Telomere length | https://gwas.mrcieu.ac.uk/datasets/ieu-b-4879/ | IEU OpenGWAS, ieu-b-4879 |
| Ben Elsworth | 2018 | Skin colour | https://gwas.mrcieu.ac.uk/datasets/ukb-b-19560/ | IEU OpenGWAS, ukb-b-19560 |

*Continued*

| Author(s) | Year | Dataset title | Dataset URL | Database and Identifier |
|-----------|------|---------------|-------------|-------------------------|
| Ben Elsworth | 2018 | Ease of skin tanning | https://gwas.mrcieu.ac.uk/datasets/ukb-b-533/ | IEU OpenGWAS, ukb-b-533 |
| Ben Elsworth | 2018 | Pack years of smoking | https://gwas.mrcieu.ac.uk/datasets/ukb-b-10831/ | IEU OpenGWAS, ukb-b-10831 |
| Ben Elsworth | 2018 | Time spent doing vigorous physical activity | https://gwas.mrcieu.ac.uk/datasets/ukb-b-13702/ | IEU OpenGWAS, ukb-b-13702 |
| Ben Elsworth | 2018 | Age completed full time education | https://gwas.mrcieu.ac.uk/datasets/ukb-b-6134/ | IEU OpenGWAS, ukb-b-6134 |
| Lee, et al | 2018 | Years of schooling | https://gwas.mrcieu.ac.uk/datasets/ieu-a-1239/ | IEU OpenGWAS, ieu-a-1239 |
| Ben Elsworth | 2018 | Alcohol intake frequency | https://gwas.mrcieu.ac.uk/datasets/ukb-b-5779/ | IEU OpenGWAS, ukb-b-5779 |
| Locke, et al | 2015 | Body mass index | https://gwas.mrcieu.ac.uk/datasets/ieu-a-835/ | IEU OpenGWAS, ieu-a-835 |
| Shungin, et al | 2015 | Waist circumference | https://gwas.mrcieu.ac.uk/datasets/ieu-a-61/ | IEU OpenGWAS, ieu-a-61 |

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

# Appendix 1

## Additional Methods

### Cancer datasets

### UK Biobank

The UK Biobank is a large cohort study including around 500,000 individuals aged 40–69 years at the time of recruitment (2006–2010). The cohort has been described in detail in previous publications (*Sudlow et al., 2015*; *Bycroft et al., 2018*). In short, all participants provided written informed consent, after which baseline data were collected using sociodemographic, lifestyle and health-related questionnaires, physical and cognitive assessments, and biological samples. Participants' data were linked to their health records for longitudinal follow-up. The study obtained ethical approval from the National Information Governance Board for Health and Social Care and the North-West Multicenter Research Ethics Committee (Ref: 11/NW/0382).

Cancer cases (diagnosed prior or after enrolment) were obtained from the UK Cancer Registry (updated to April 2019). They were then coded according to the ninth and tenth editions of the International Classification of Diseases (ICD-9 and ICD-10, respectively) as follows: breast (ICD-9: 174; ICD-10: C50), ovarian (ICD-9: 183; ICD-10: C56), prostate (ICD-9: 185; ICD-10: C61), lung (ICD-9: 162; ICD-10: C34) and colorectal cancer (ICD-9: 153; ICD-10: C18-C20). Controls excluded individuals with any type of cancer (self-reported and/or recorded in cancer registry), as well as those with benign, in situ, uncertain or unspecified behaviour neoplasms (ICD-9: 210–239; ICD-10: D00-D49).

Sample-level quality control (QC) involved removing any individuals who had non-white British genetic ancestry, sex chromosome aneuploidies, who withdrew consent from the UK Biobank study and who were closely related to other participants. Variant-level QC consisted in imputing SNPs using the Haplotype Reference Consortium (HRC) and restricting SNPs to a minor allele frequency (MAF) > 0.1%, a genotyping rate > 0.015 and a Hardy-Weinberg Equilibrium (HWE) $P > 1 \times 10^{-4}$. LD pruning was performed to an $r^2$ cutoff of 0.1 using PLINK v2 (*Mitchell et al., 2019*). In order to reduce false positive signals, SNPs were removed when MAF was below our expectations (we would expect at least 25 minor alleles in cases), as recommended in http://www.nealelab.is/blog/2017/9/11/details-and-considerations-of-the-uk-biobank-gwas.

The GWAS analysis in the UK Biobank consisted of 13,879 cases and 198,523 controls for breast cancer, 1,218 cases and 198,523 controls for ovarian cancer, 9,132 cases and 173,493 controls for prostate cancer, 2,671 cases and 372,016 controls for lung cancer and 5,657 cases and 372,016 controls for colorectal cancer. It was performed using BOLT-LMM v2.3.5 (*Loh et al., 2015*; *Elsworth et al., 2019*), adjusting for sex and genotyping chip. BOLT-LMM uses a linear mixed model to account for population stratification and cryptic relatedness in the UK Biobank. Lung cancer associations were estimated twice, once adjusting for genotyping chip and once without. Since UKBiLEVE participants were genotyped using a different array and using adjusted lung cancer estimates may introduce collider bias, we only included MR results obtained using the unadjusted lung cancer GWAS estimates in the meta-analysis. For sex-specific cancers, analyses were limited to individuals of the pertinent sex (only females were used for breast and ovarian cancers, whereas only males were used for prostate cancer). Beta coefficients and their corresponding standard errors were finally transformed to log odds ratios (ORs) (*Elsworth et al., 2019*).

We also performed a GWAS analysis of parental history of cancer reported by UK Biobank participants (i.e. breast, prostate, lung and bowel cancer) using BOLT-LMM software v2.3.5 (*Loh et al., 2015*). Age and sex were included as covariates in the model as before. For sex-specific cancers, analyses were restricted to individuals of the relevant sex (i.e. maternal history only for breast cancer and paternal history only for prostate cancer). We obtained 35,356 breast cancer cases and 206,992 controls, in addition to 31,527 prostate cancer cases and 160,579 controls. For other cancers, we combined maternal and paternal history of cancer, thus obtaining a total of 51,073 lung cancer cases and 404,606 controls, as well as 45,213 bowel cancer cases and 412,429 controls. GWAS of these outcomes have previously provided strong concordance with those based on hospital records (*DeBoever et al., 2020*). They have also provided consistent results in MR (*Richardson et al., 2021*).

## FinnGen

The FinnGen R5 release includes data on 218,792 individuals of Finnish ancestry, obtained from Finnish biobanks and digital health registry records (**FinnGen, 2021**). Complete study details are available elsewhere (https://www.finngen.fi/en). In brief, samples were excluded for the following reasons: ambiguous gender, genotype missingness > 5%, heterozygosity +–4 s.d. and non-Finnish ancestry. SNPs were genotyped using Illumina and Affymetrix arrays. Variants were excluded for the following reasons: missingness > 2%, HWE $P < 1 \times 10\text{-}6$ and minor allele count < 3. Genotypes were imputed using the Finnish SISu v3 reference panel. The GWAS analysis was conducted using SAIGE v0.36.3.2, a mixed model logistic regression R/C ++ package. Sex, age, genotyping batch and the first 10 genetically derived principal components were included as covariates in the analysis. We used FinnGen R5 release data on breast (8,401 cases and 99,321 controls), ovarian (719 cases and 99,321 controls), prostate (6,311 cases and 74,685 controls), lung (1,681 cases and 173,933 controls) and colorectal cancer (3,022 cases and 174,006 controls). We used the "EXALLC" cancer variables, which excluded other cancers from controls.

## Breast Cancer Association Consortium

The GWAS summary data for breast cancer were obtained from a Breast Cancer Association Consortium (BCAC) meta-analysis performed by **Michailidou et al., 2017**. This included 122,977 cases and 105,974 controls (69,501 cases of ER + and 21,468 of ER- breast cancer). All studies that contributed to this meta-analysis have been fully detailed in previous publications (**Michailidou et al., 2017**; **Michailidou et al., 2013**; **Michailidou et al., 2015**). In sum, samples were excluded if they had a low call rate ( < 95%), abnormally high or low heterozygosity (4.89 s.d. from the mean), < 80% European ancestry, probable duplicates and/or close relatives within and across studies. Genetic variants were genotyped using the Illumina OncoArray and iCOGS arrays and genotypes were imputed using the 1,000 Genomes Project Phase three reference panel. The GWAS analysis was performed using logistic regression models, adjusting for up to 10 principal components and either country or study. This was done using purpose-written software. OncoArray and iCOGS estimates were combined in a fixed-effect inverse variance weighted meta-analysis using the METAL software (**Willer et al., 2010**). Only SNPs with r2 ≥0.3 and MAF ≥0.005 were included in the meta-analysis.

We also obtained summary data for breast cancer subtypes from a BCAC GWAS meta-analysis by **Zhang et al., 2020**. The study comprised data on luminal A-like (7,325 cases), luminal B-like (1,682 cases), luminal B/HER2-negative-like (1,779 cases), HER2-enriched-like (718 cases) and triple-negative (2,006 cases) invasive breast cancer subtypes and 20,815 controls. The details of the study can be found in the publication. In brief, the analyses excluded cases of carcinoma in situ, cases missing data on tumour characteristics and cases for which there were no controls available in their respective countries. Participants were also excluded if age at diagnosis/enrolment was missing. Genotypes were obtained using OncoArray and iCOGS arrays. Imputation was performed using the 1,000 Genomes Project Phase three reference panel. OncoArray and iCOGS datasets were analysed separately using two-stage polytomous logistic regression analyses in R. Models were adjusted for age and the first 10 principal components. SNPs with r2 < 0.3 and MAF < 0.01 were excluded from the subtype analyses, as well as those in linkage disequilibrium (r2 ≥ 0.1) or within ± 500 kb of known susceptibility SNPs. GWAS results were then pooled using fixed-effect meta-analysis in METAL (**Willer et al., 2010**).

## Ovarian Cancer Association Consortium

We used ovarian cancer genetic summary statistics from an Ovarian Cancer Association Consortium (OCAC) study by **Phelan et al., 2017**. This comprised 25,509 cases and 40,941 controls. Subtypes included high grade serous (13,037 cases), low grade serous (1,012 cases), invasive mucinous (1,417 cases), clear cell (1,366 cases) and endometrioid (2,810 cases) ovarian cancers. This study combined genotype data from OCAC and Consortium of Investigators of Modifiers of BRCA1/2 (CIMBA) genotyping projects. These have been fully described in the publication. In short, samples with > 27% non-European ancestry were excluded, as were those with a genotyping call rate < 95%, excessively low or high heterozygosity. Non-females and duplicates were also removed. SNPs were genotyped using several Illumina arrays (OncoArray, iSelect iCOGS, 550 k, HumanOmni 2.5 M, 610 Quad and 317 k). Imputations were performed separately for each genotyping project using the 1,000 Genomes Project v3 reference panel. GWAS analyses were conducted in custom-written software using logistic regression models adjusted for study and principal components. SNPs with r2 < 0.3 and MAF < 0.01 were excluded. GWAS estimates were pooled using fixed effect meta-analysis in METAL (**Willer et al., 2010**).

## Consortium of Investigators of Modifiers of BRCA1/2

We also used CIMBA GWAS data for breast and ovarian cancers in BRCA1 and BRCA2 mutation carriers (*Phelan et al., 2017*; *Milne et al., 2017*). The genotyping and imputation procedures that were used have been described elsewhere (*Phelan et al., 2017*; *Milne et al., 2017*). In brief, samples were excluded if they were non-female, had discordant genotypes in known sample duplicates, had > 19% non-European ancestry, a genotyping call rate < 95% or extremely low or high heterozygosity ($P < 1 \times 10^{-6}$). SNPs were genotyped using Illumina's Oncoarray and iSelect Collaborative Oncological Gene-Environment Study (iCOGS) arrays. Imputation was performed using the 1,000 Genomes Project Phase three reference panel. GWAS analyses were conducted separately for BRCA1 and BRCA2 mutation carriers and for iCOGS and OncoArray samples. Genetic association data was generated using a survival analysis framework, using a retrospective likelihood approach. Analyses were stratified by country of origin and Ashkenazi Jewish origin. Custom written functions in Fortran and Python were used to carry out the analyses and kinship-adjusted score test statistics were implemented in R software. OncoArray and iCOGS results were pooled using fixed-effect meta-analysis in METAL (*Willer et al., 2010*).

## Prostate Cancer Association Group to Investigate Cancer Associated Alterations in the Genome

Prostate cancer GWAS summary data were acquired from a Prostate Cancer Association Group to Investigate Cancer Associated Alterations in the Genome (PRACTICAL) study by *Schumacher et al., 2018*. This included 79,148 cases and 61,106 controls. It also comprised data on prostate cancer subtypes: 15,167 advanced cases vs. 58,308 healthy controls; 14,160 advanced cases vs. 62,421 non-advanced controls; 6,988 early-onset cases (age at diagnosis ≤ 55 years) vs. 44,256 healthy controls; 15,561 high aggressive cases vs. 9,739 low aggressive controls; and 20,658 high aggressive cases vs. 38,093 low/intermediate aggressive controls.

Prostate cancer aggressiveness was defined as follows:

- Low aggressive: tumor stage ≤T1 **AND** Gleason score ≤6 **AND** prostate-specific antigen (PSA) <10 ng/mL.
- Intermediate aggressive: tumor stage T2 **OR** Gleason score = 7 **OR** PSA 10–20 ng/mL.
- High aggressive: tumor stage T3/T4, N1 or M1 **OR** Gleason score ≥8 **OR** PSA >20 ng/mL.
- Advanced: metastatic disease **OR** Gleason score ≥ 8 **OR** PSA > 100 ng/mL **OR** death due to prostate cancer.

Study details are available in the publication. In brief, individuals were excluded if they presented a call rate < 95%, extreme heterozygosity ( > 4.9 s.d. from the mean), if they were duplicates or if they were related to other participants. Only men of European ancestry ( > 80%) were included in the GWAS. Studies were genotyped using Illumina (OncoArray, Human 610, 60 k, Infinium HumanHap 550, iSELECT, iCOGS and Human Omni 2.5) and Affymetrix GeneChip (500 k and 5.0 k) genotyping arrays and SNPs were imputed to the 1,000 Genomes Project Phase three reference panel. Genetic association data were obtained using logistic regression analysis. Models were adjusted for seven principal components and study-relevant covariates and stratified by country or study. Odds ratios were derived using either SNPTEST or a custom written C ++ software. GWAS estimates were combined using fixed-effect meta-analysis in METAL (*Willer et al., 2010*).

## International Lung Cancer Consortium

For lung cancer, we used genetic summary data obtained from an International Lung Cancer Consortium (ILCCO) GWAS meta-analysis of 11,348 cases and 15,861 controls by *Wang et al., 2014*. We also used lung cancer subtype data including 3,275 squamous cell lung carcinoma cases and 15,038 controls, as well as 3,442 lung adenocarcinoma cases and 14,894 controls. Individual studies included in the meta-analysis have been explained in prior publications (*Timofeeva et al., 2012*; *Amos et al., 2008*; *Wang et al., 2008*; *Hung et al., 2008*). In summary, sample QC consisted in excluding any individuals of non-European ancestry, with low call rates ( < 90%), or abnormally high or low heterozygosity ($P < 1 \times 10^{-4}$). Duplicates and closely related individuals were also removed. Genotyping was performed using Illumina HumanHap 317 k, 317k + 240 S, 370Duo, 550 k, 610 k or 1 M arrays. SNPs were imputed from the 1,000 Genomes Project Phase 1 v3 reference panel. GWAS estimates were obtained by unconditional logistic regression in R v2.6, Stata v.10 and PLINK v1.06 software. Analyses were adjusted for principal components. Fixed-effect meta-analysis was used to pool estimates across studies.

## Genetics and Epidemiology of Colorectal Cancer Consortium

Colorectal cancer GWAS summary statistics were retrieved from a Genetics and Epidemiology of Colorectal Cancer Consortium (GECCO) GWAS meta-analysis by *Huyghe et al., 2019*. This comprised 58,131 cases (31,288 male and 26,843 female) and 67,347 controls (34,527 male and 32,820 female). Cases were defined as patients with colorectal cancer or advanced adenoma.

Data on colorectal cancer subtypes were obtained from another GECCO publication by *Huyghe et al., 2021*. This included 48,214 cases and 64,159 controls (32,002 colon, 15,706 proximal colon, 14,376 distal colon and 16,212 rectal cancer cases).

Colorectal cancer subtypes were defined as follows:

- Proximal colon cancer: any primary tumour starting in the cecum, ascending colon, hepatic flexure, or transverse colon (ICD-9: 153.4, 153.6, 153.0, or 153.1, respectively).
- Distal colon cancer: any primary tumour starting in the splenic flexure, descending colon, or sigmoid colon (ICD-9 codes: 153.7, 153.2, or 153.3, respectively)
- Colon cancer: proximal and distal colon cancer cases, in addition to colon cancer cases with unspecified site.
- Rectal cancer: any primary tumour starting in the rectum or rectosigmoid junction (ICD-9 codes: 154.1, or 154.0, respectively)

Controls excluded individuals with known history of cancer or reported family history of colorectal cancer. QC procedures have been explained in the publications (*Huyghe et al., 2019*; *Huyghe et al., 2021*). In brief, the studies excluded samples with evidence of DNA contamination, high missing genotype rates, unintentional duplicate pairs and sex discrepancies. Closely related individuals and those of non-European ancestry were also excluded. Genotyping was conducted using Illumina (300 k, Oncoarray, 1 M, 550 k, 610 k, OmniExpress, OmniExpressExome, 300/240 S and custom iSelect) and Affymetrix (Axiom and 500 k) arrays. Imputation was performed to the HRC reference panel. GWAS analyses were conducted for SNPs with an imputation accuracy r2 ≥ 0.3 and minor allele count ≥ 50 using logistic regression models adjusted for principal components, age, sex and study-specific covariates. METAL (*Willer et al., 2010*) was used to combine summary statistics across studies using fixed-effect meta-analysis.

## Sensitivity analyses

MR-Egger assumes that the association between SNPs and epigenetic age acceleration is not correlated with SNPs that affect cancer via pleiotropic pathways (Instrument Strength Independent of Direct Effect—InSIDE assumption) (*Bowden et al., 2015*). The weighted median method assumes that at least half of the SNPs in the analysis are valid instruments. The weighted mode approach presupposes that the most frequent association estimate is not affected by pleiotropy, meaning it must correspond to the true causal effect (ZEro Modal Pleiotropy Assumption—ZEMPA) (*Hartwig et al., 2017*).

## Data availability

Summary statistics for epigenetic age acceleration measures of HannumAge, Intrinsic HorvathAge, PhenoAge and GrimAge were downloaded from: https://datashareed.ac.uk/handle/10283/3645. Summary statistics for international cancer genetic consortiums were obtained from their respective data repositories. Colorectal cancer data were obtained following the submission of a written request to the GECCO committee, which may be contacted by email at kafdem@fredhutch.org/upeters@fredhutch.org. Breast, ovarian, prostate and lung cancer data were accessed via MR-Base (http://app.mrbase.org/), which holds complete GWAS summary data from BCAC, OCAC, PRACTICAL and ILCCO. Breast cancer subtype data were obtained from BCAC and can be downloaded from: http://bcacccgemedschlcam.ac.uk/bcacdata/oncoarray/oncoarray-and-combined-summary-result/gwas-summary-associations-breast-cancer-risk-2020/. Data on breast and ovarian cancer in BRCA1 and BRCA2 carriers were obtained from CIMBA and can be downloaded from: http://cimbaccgemedschlcam.ac.uk/oncoarray-complete-summary-results/. Prostate cancer subtype data are not publicly available through MR-Base but can be accessed upon request. These data are managed by the PRACTICAL committee, which may be contacted by email at practical@icr.ac.uk. FinnGen data is publicly available and can be accessed here: https://www.finngen.fi/en/access_results. UK Biobank data can be accessed through the MR-Base platform. Parental history of cancer data were obtained from the UK Biobank study under application #15825 and can be accessed via an approved application to the UK Biobank (https://www.ukbiobank.ac.uk/enable-your-research/apply-for-access). GWAS data for negative control outcomes and potential confounders were obtained via the MR-Base platform (GWAS IDs for negative control outcomes: "ukb-b-19560", "ukb-b-533";

GWAS IDs for confounders: "ukb-b-10831", "ukb-b-13702", "ukb-b-6134", "ieu-a-1239", "ukb-b-5779", "ieu-a-835", "ieu-a-61"). GWAS data for measured telomere length used in bidirectional MR analyses were also obtained via the MR-Base platform (GWAS ID: "ieu-b-4879").

## Code availability

The GWAS analysis for cancers in UK Biobank was performed using BOLT-LMM v2.3.5 (http://data.broadinstitute.org/alkesgroup/BOLT-LMM/). All MR analyses and visualisations were conducted using R software v4.0.2 (https://www.r-project.org/). For cancer datasets that were not obtained from the MR-Base platform, LD proxies were identified using the 'LDlinkR' v1.1.2 R package (https://github.com/CBIIT/LDlinkR; *Myers, 2021*). Two-sample MR analyses were conducted using the 'TwoSampleMR' v0.5.6 R package (https://github.com/MRCIEU/TwoSampleMR; *Parker, 2021*). Meta-analyses were performed using the 'meta' v4.18 R package (https://github.com/guido-s/meta/; *Schwarzer, 2022*). GWAS meta-analyses used to perform single-SNP MR analyses were done using the METAL software (https://genome.sph.umich.edu/wiki/METAL_Documentation). MR-CAUSE analyses were conducted using the 'cause' v1.2.0 R package (https://github.com/jean997/cause; *Morrison, 2021*). Plots were created using the 'ggforestplot' v0.1.0 R package (https://github.com/nightingalehealth/ggforestplot; *Jagerroos, 2020*). LD scores were computed using the 'ldsc' v1.0.1 command line tool (https://github.com/bulik/ldsc; *Schorsch, 2020*). The code used in this study is available at: https://github.com/fernandam93/epiclocks_cancer; *Morales Berstein, 2021*.

## Appendix 2

## Additional Figures

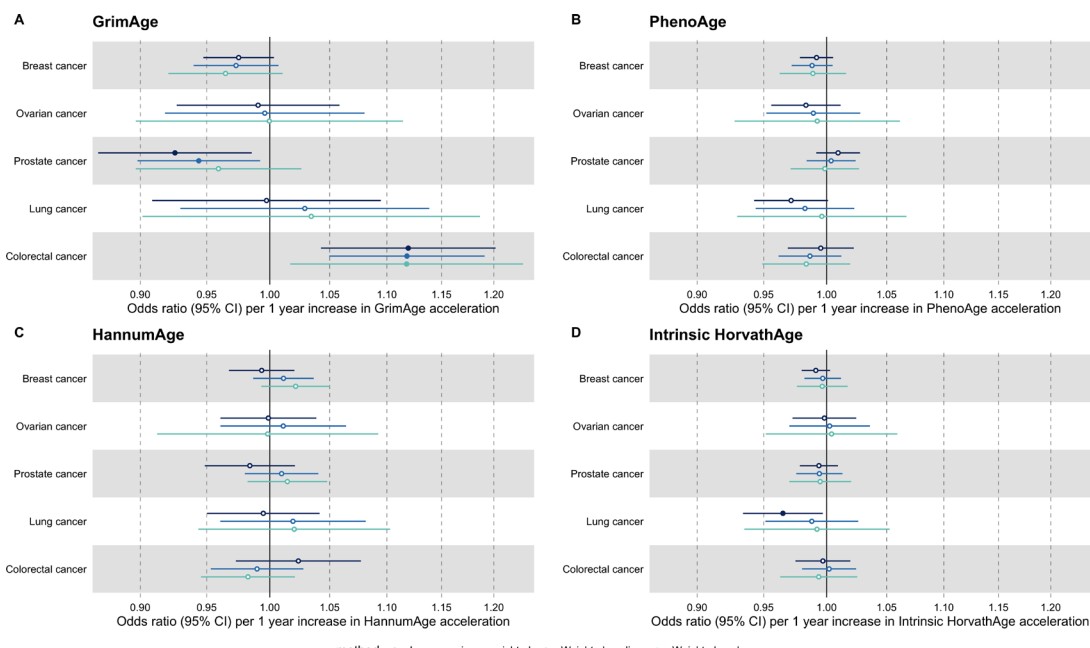

**Appendix 2—figure 1.** Fixed effect meta-analysis of Mendelian randomization estimates for genetically predicted effects of epigenetic age acceleration on multiple cancers. Odds ratios and 95% confidence intervals are reported per 1 year increase in (**A**) GrimAge acceleration, (**B**) PhenoAge acceleration, (**C**) HannumAge acceleration and (**D**) Intrinsic HorvathAge acceleration. Results were obtained using inverse variance weighted MR (dark blue), weighted median (sky blue) and weighted mode (turquoise) methods. All meta-analysis estimates were calculated using data from UK Biobank, FinnGen and international consortia, except for colorectal cancer estimates, which exclude UK Biobank data to avoid double counting.

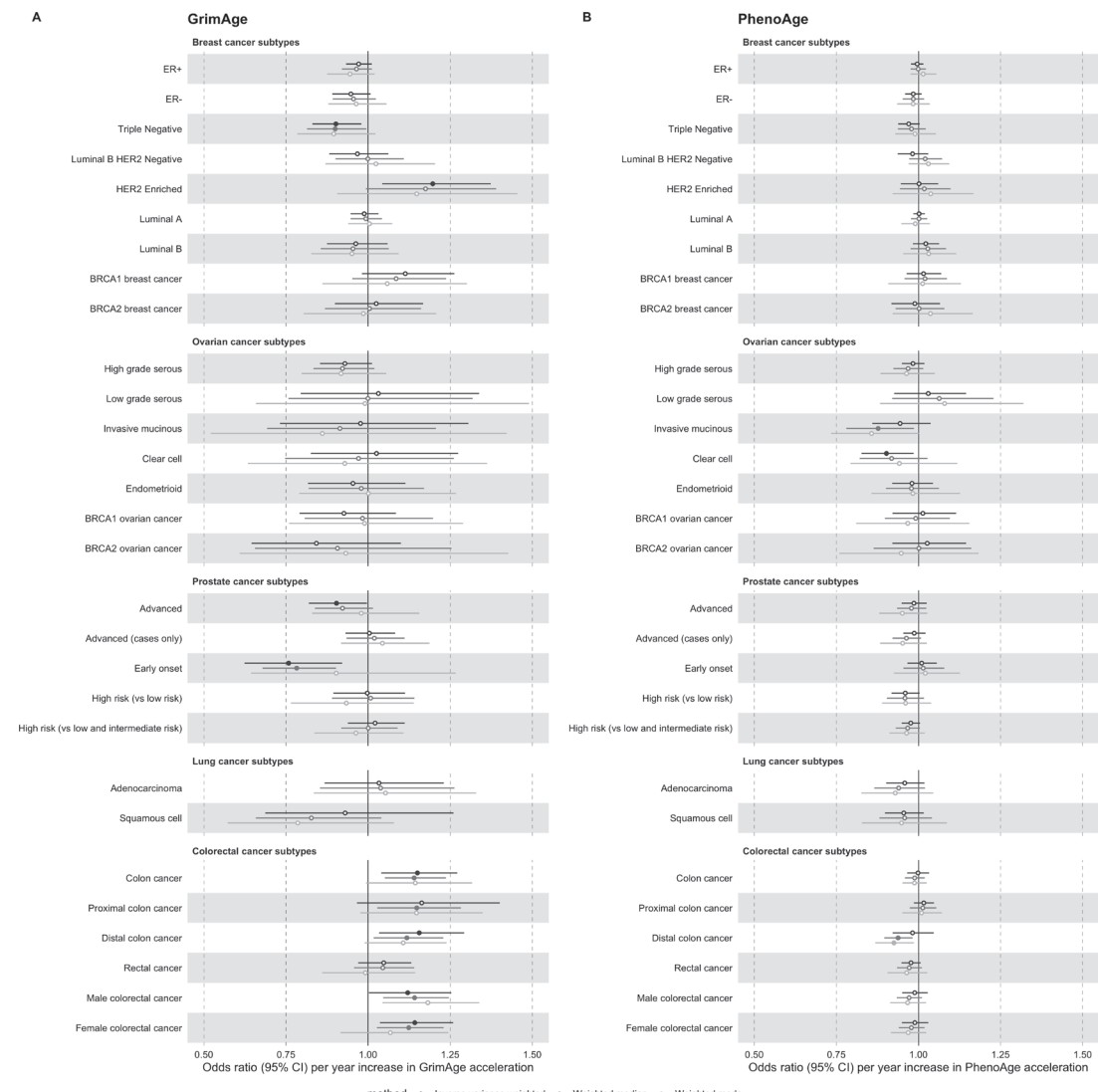

**Appendix 2—figure 2.** Mendelian randomization estimates for genetically predicted effects of GrimAge and PhenoAge acceleration on multiple cancer subtypes. Odds ratios and 95% confidence intervals are reported per 1 year increase in (**A**) GrimAge acceleration, (**B**) PhenoAge acceleration. GrimAge and PhenoAge acceleration were instrumented by four and 11 genetic variants, respectively. Results were obtained using inverse variance weighted MR (dark blue), weighted median (sky blue) and weighted mode (turquoise) methods. Data sources: BCAC, OCAC, CIMBA, PRACTICAL, ILCCO and GECCO.

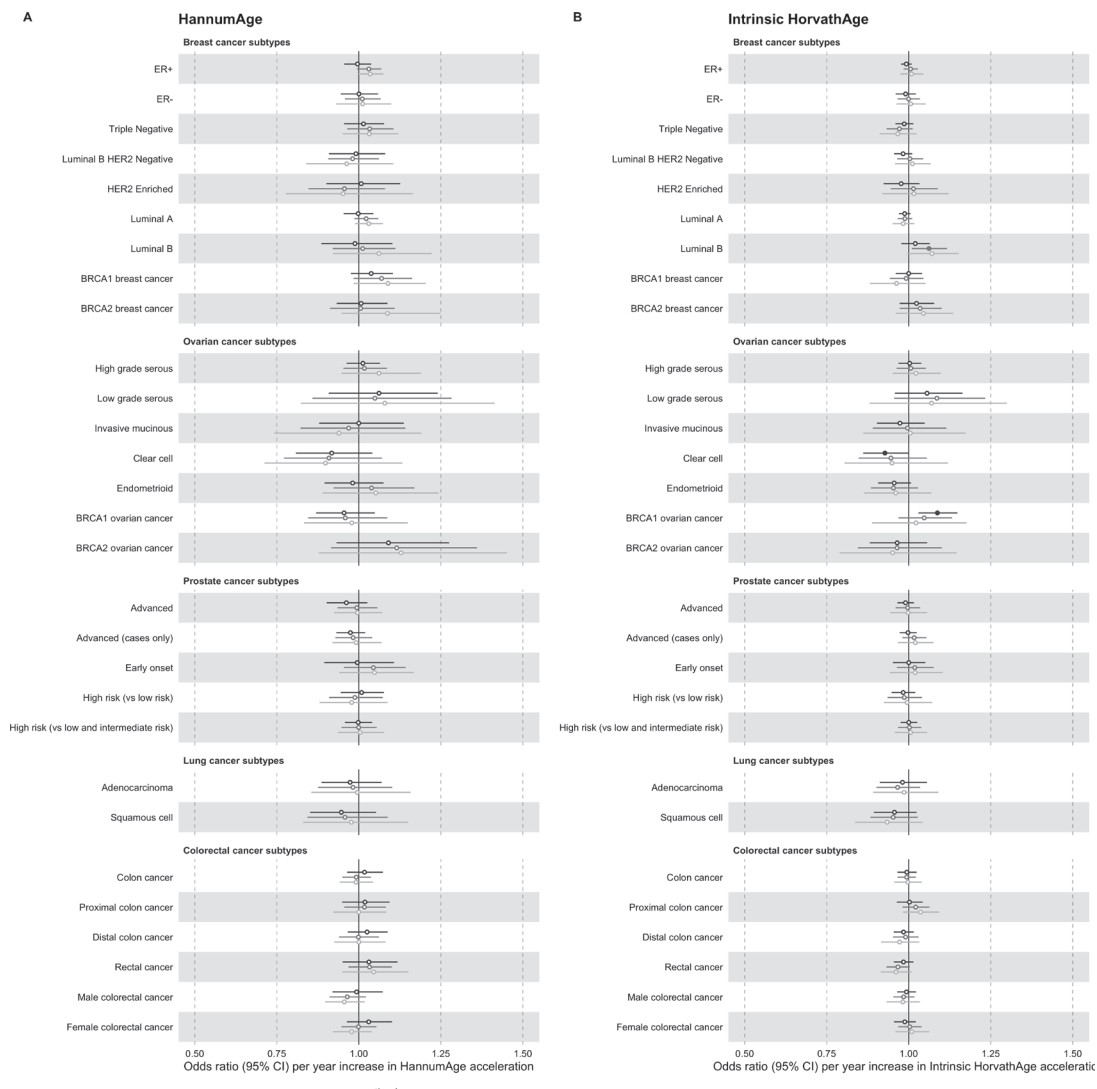

**Appendix 2—figure 3.** Mendelian randomization estimates for genetically predicted effects of HannumAge and Intrinsic HorvathAge acceleration on multiple cancer subtypes. Odds ratios and 95% confidence intervals are reported per 1 year increase in (**A**) HannumAge acceleration, (**B**) Intrinsic HorvathAge acceleration. HannumAge and Intrinsic HorvathAge acceleration were instrumented by nine and 24 genetic variants, respectively. Results were obtained using inverse variance weighted MR (dark blue), weighted median (sky blue) and weighted mode (turquoise) methods. Data sources: BCAC, OCAC, CIMBA, PRACTICAL, ILCCO and GECCO.

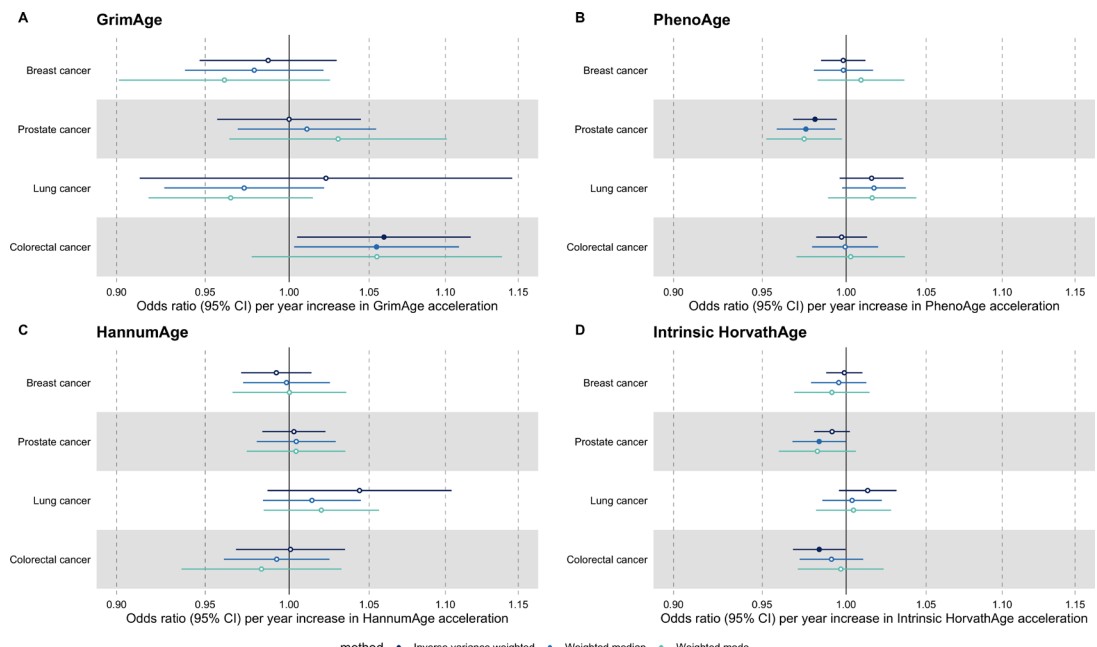

**Appendix 2—figure 4.** Mendelian randomization estimates for genetically predicted effects of epigenetic age acceleration on parental history of multiple cancers. Odds ratios and 95% confidence intervals are reported per 1 year increase in (**A**) GrimAge acceleration, (**B**) PhenoAge acceleration, (**C**) HannumAge acceleration and (**D**) Intrinsic HorvathAge acceleration. GrimAge, PhenoAge, HannumAge and Intrinsic HorvathAge acceleration were instrumented by 4, 11, 9 and 24 genetic variants, respectively. Results were obtained using inverse variance weighted MR (dark blue), weighted median (sky blue) and weighted mode (turquoise) methods. Data source: UK Biobank.

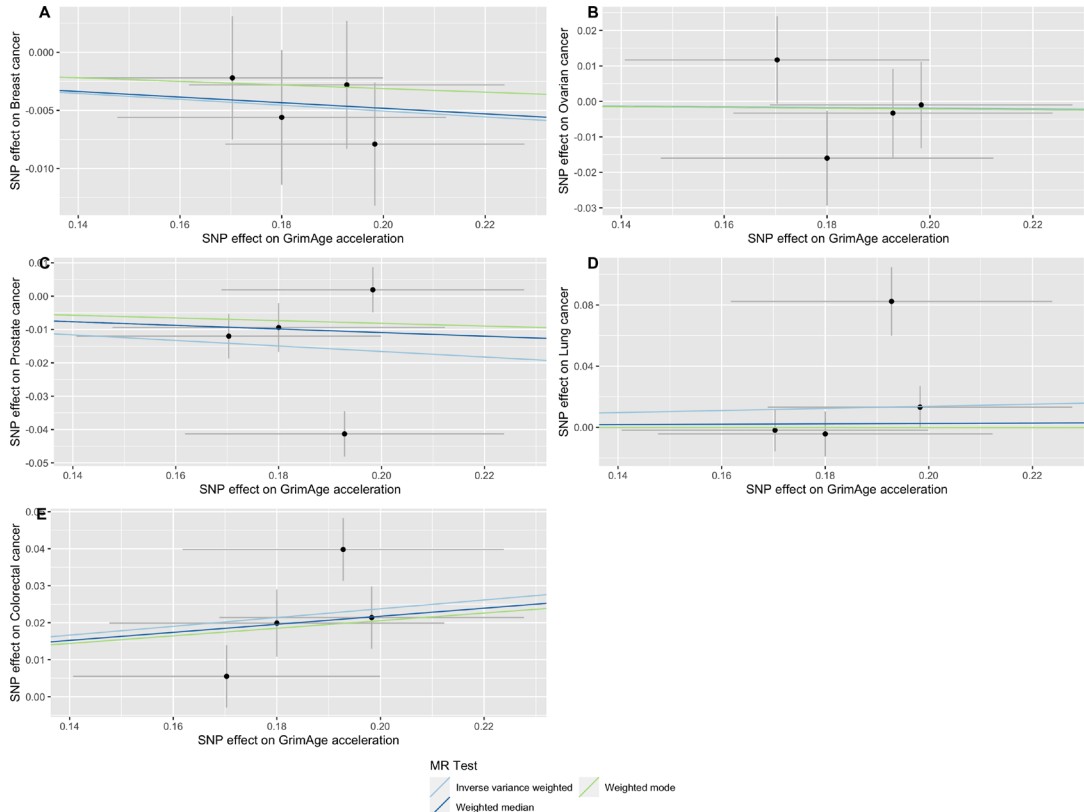

**Appendix 2—figure 5.** Scatter plot showing the effect of genetic instruments on GrimAge acceleration against their effect on multiple cancer. Genome-wide association estimates for (**A**) breast, (**B**) ovarian, (**C**) prostate, (**D**) lung and (**E**) colorectal cancer were meta-analysed using the METAL software. Results were obtained using inverse variance weighted MR (light blue), weighted median (dark blue) and weighted mode (light green) methods. Data sources: UK Biobank, FinnGen and international cancer genetic consortia. For colorectal cancer, UK Biobank estimates were not included in the meta-analysis to avoid double counting participants included in the GECCO consortium.

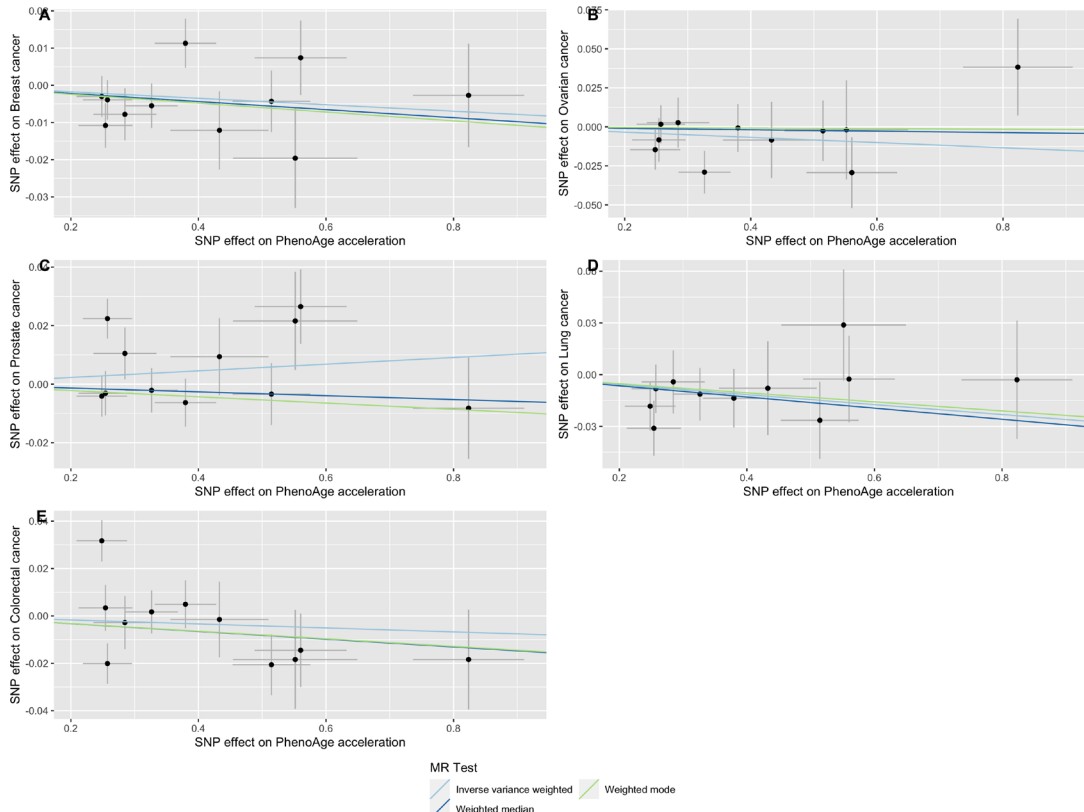

**Appendix 2—figure 6.** Scatter plot showing the effect of genetic instruments on PhenoAge acceleration against their effect on multiple cancer. Genome-wide association estimates for (**A**) breast, (**B**) ovarian, (**C**) prostate, (**D**) lung and (**E**) colorectal cancer were meta-analysed using the METAL software. Results were obtained using inverse variance weighted MR (light blue), weighted median (dark blue) and weighted mode (light green) methods. Data sources: UK Biobank, FinnGen and international cancer genetic consortia. For colorectal cancer, UK Biobank estimates were not included in the meta-analysis to avoid double counting participants included in the GECCO consortium.

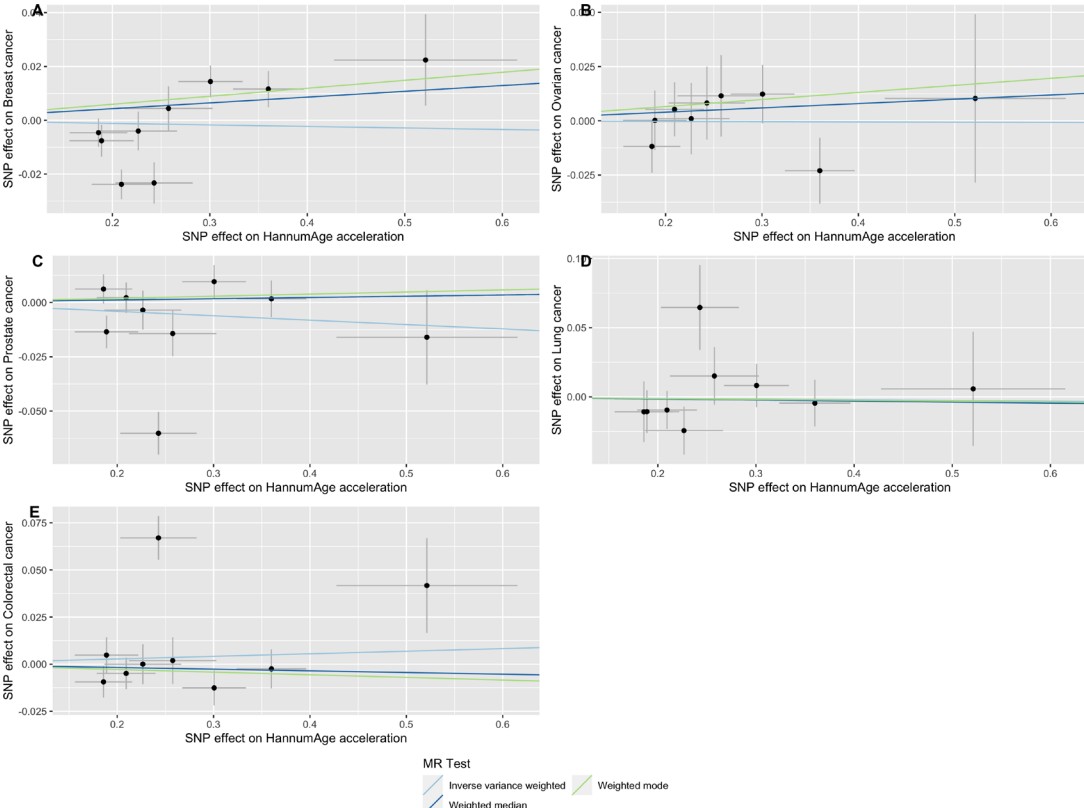

**Appendix 2—figure 7.** Scatter plot showing the effect of genetic instruments on HannumAge acceleration against their effect on multiple cancer. Genome-wide association estimates for (**A**) breast, (**B**) ovarian, (**C**) prostate, (**D**) lung and (**E**) colorectal cancer were meta-analysed using the METAL software. Results were obtained using inverse variance weighted MR (light blue), weighted median (dark blue) and weighted mode (light green) methods. Data sources: UK Biobank, FinnGen and international cancer genetic consortia. For colorectal cancer, UK Biobank estimates were not included in the meta-analysis to avoid double counting participants included in the GECCO consortium.

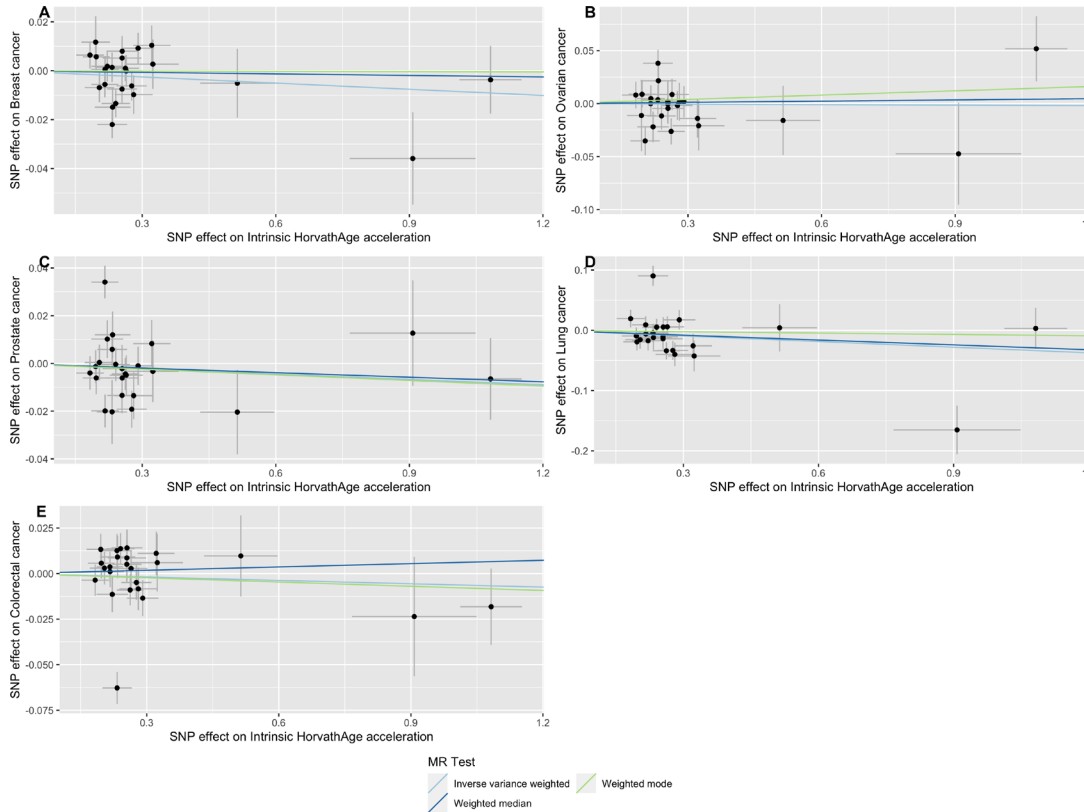

**Appendix 2—figure 8.** Scatter plot showing the effect of genetic instruments on Intrinsic HorvathAge acceleration against their effect on multiple cancer. Genome-wide association estimates for (**A**) breast, (**B**) ovarian, (**C**) prostate, (**D**) lung and (**E**) colorectal cancer were meta-analysed using the METAL software. Results were obtained using inverse variance weighted MR (light blue), weighted median (dark blue) and weighted mode (light green) methods. Data sources: UK Biobank, FinnGen and international cancer genetic consortia. For colorectal cancer, UK Biobank estimates were not included in the meta-analysis to avoid double counting participants included in the GECCO consortium.

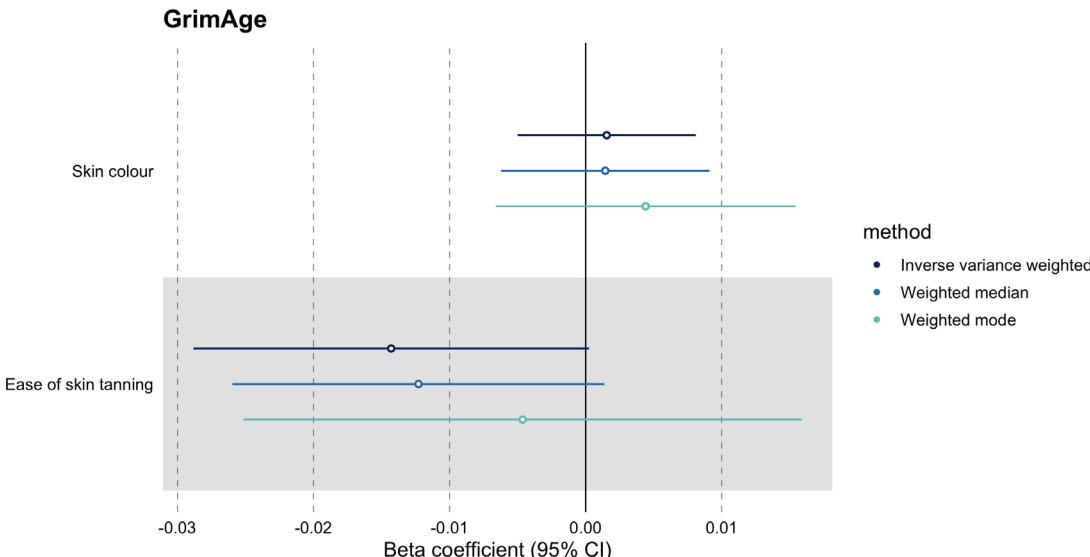

**Appendix 2—figure 9.** Mendelian randomization estimates for genetically predicted effects of GrimAge acceleration on negative control outcomes. Odds ratios and 95% confidence intervals are reported per 1 year increase in GrimAge acceleration. GrimAge was instrumented by four genetic variants. Results were obtained using inverse variance weighted MR (dark blue), weighted median (sky blue) and weighted mode (turquoise) methods. Data source: UK Biobank.

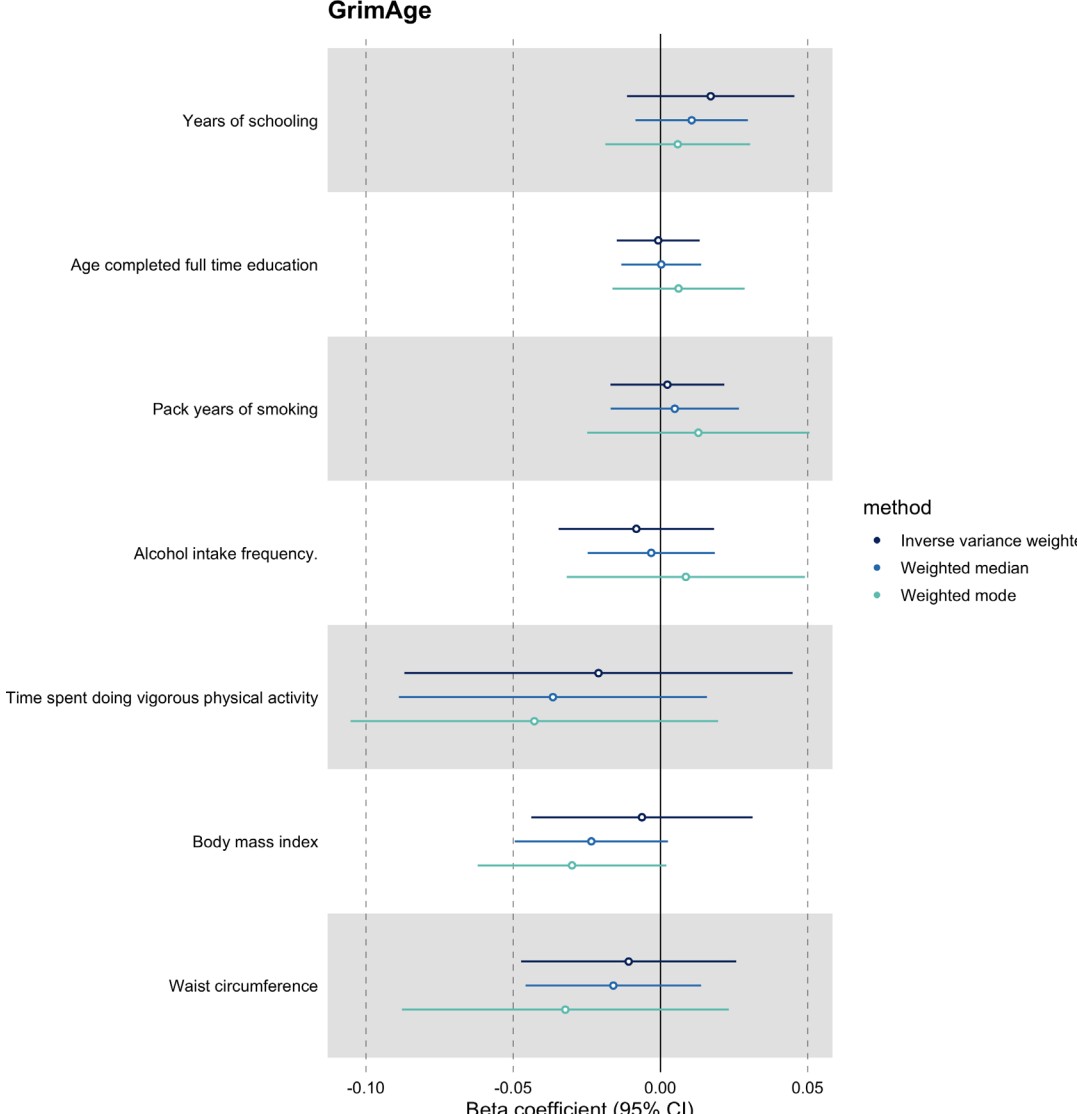

**Appendix 2—figure 10.** Mendelian randomization estimates for genetically predicted effects of GrimAge acceleration on potential confounders of the association between GrimAge acceleration and colorectal cancer. Odds ratios and 95% confidence intervals are reported per 1 year increase in GrimAge acceleration. GrimAge was instrumented by four genetic variants. Results were obtained using inverse variance weighted MR (dark blue), weighted median (sky blue) and weighted mode (turquoise) methods. Data sources: UK Biobank (for time spent doing vigorous physical activity, pack years of smoking, alcohol intake frequency and age completed full time education), GIANT consortium (for waist circumference and body mass index) and the SSAGC consortium (for years of schooling).

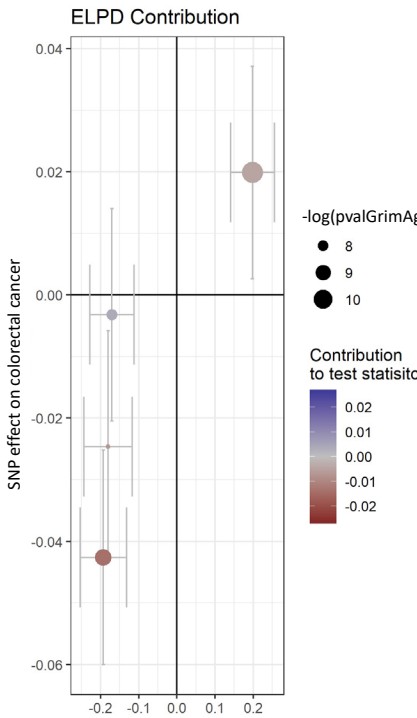

| | model1 | model2 | delta_elpd | se_delta_elpd | z | p |
|---|---|---|---|---|---|---|
| 1 | null | sharing | 0.37 | 0.03 | 12.0 | 1 |
| 2 | null | causal | 1.10 | 0.14 | 8.3 | 1 |
| 3 | sharing | causal | 0.77 | 0.12 | 6.3 | 1 |

| model | gamma | eta | q |
|---|---|---|---|
| Sharing | NA | 0 (-0.31, 0.31) | 0.04 (0, 0.24) |
| Causal | 0 (-0.04, 0.04) | 0 (-0.3, 0.31) | 0.04 (0, 0.26) |

**Appendix 2—figure 11.** CAUSE analysis for the genetically predicted effect of GrimAge acceleration on colorectal cancer in GECCO. CAUSE estimates for colorectal cancer reported per 1 year increase in GrimAge acceleration. The ELPD Contribution plot shows the relative contribution of each SNP to the CAUSE test statistic. Only SNPs with $P < 5e-8$ are shown. SNPs represented by larger circles reflect smaller p-values for the associations between genetic variants and GrimAge acceleration. SNPs that contribute more to the causal model are shown in warmer tones (i.e. red), while those that contribute more to the sharing model are shown in colder tones (i.e. blue). The delta_elpd is the statistic used to compare models. It is equal to elpd(model 1)- elpd(model 2). In the upper table, positive delta_elpd's suggest that model one is a better fit to the data than model 2 (i.e. that the null model is better than the sharing model in row 1, that the null model is better than the causal model in row 2, and that the sharing model is better than the causal model in row 3). The corresponding p-values test whether model two is a better fit than model 1. Here, row three suggests that the causal model is not a better fit than the sharing model (the delta_elpd is positive and the p-value is 1, so there is no detectable evidence against the null hypothesis that the sharing model is better than the causal model). In the bottom table, eta represents the sharing factor effect (SNPs affect shared factor and shared factor simultaneously affects GrimAge and colorectal cancer) and gamma represents the causal factor effect (SNPs affect GrimAge and GrimAge affects colorectal cancer). Here, "0 (-0.04, 0.04)" represents the genetically predicted effect of GrimAge acceleration on colorectal cancer after adjusting for correlated and uncorrelated horizontal pleiotropy (results in log odds ratio scale). The intervals shown are credible intervals. Data source: GECCO, Genetics and Epidemiology of Colorectal Cancer Consortium.

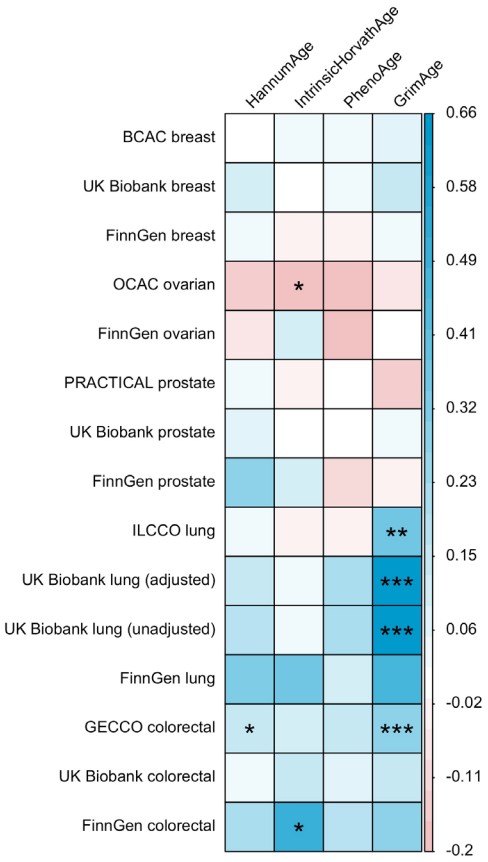

**Appendix 2—figure 12.** Genetic correlation estimates for epigenetic age acceleration and multiple cancers. Genetic correlation coefficients are reported per 1 year increase in epigenetic age acceleration. Results were obtained using LD Score regression. Abbreviations: BCAC, Breast Cancer Association Consortium; OCAC, Ovarian Cancer Association Consortium; PRACTICAL, Prostate Cancer Association Group to Investigate Cancer Associated Alterations in the Genome; ILCCO, International Lung Cancer Consortium; GECCO, Genetics and Epidemiology of Colorectal Cancer Consortium. For UK Biobank lung cancer results, adjusted means results have been adjusted for genotyping chip and unadjusted means results have not been adjusted for genotyping chip.

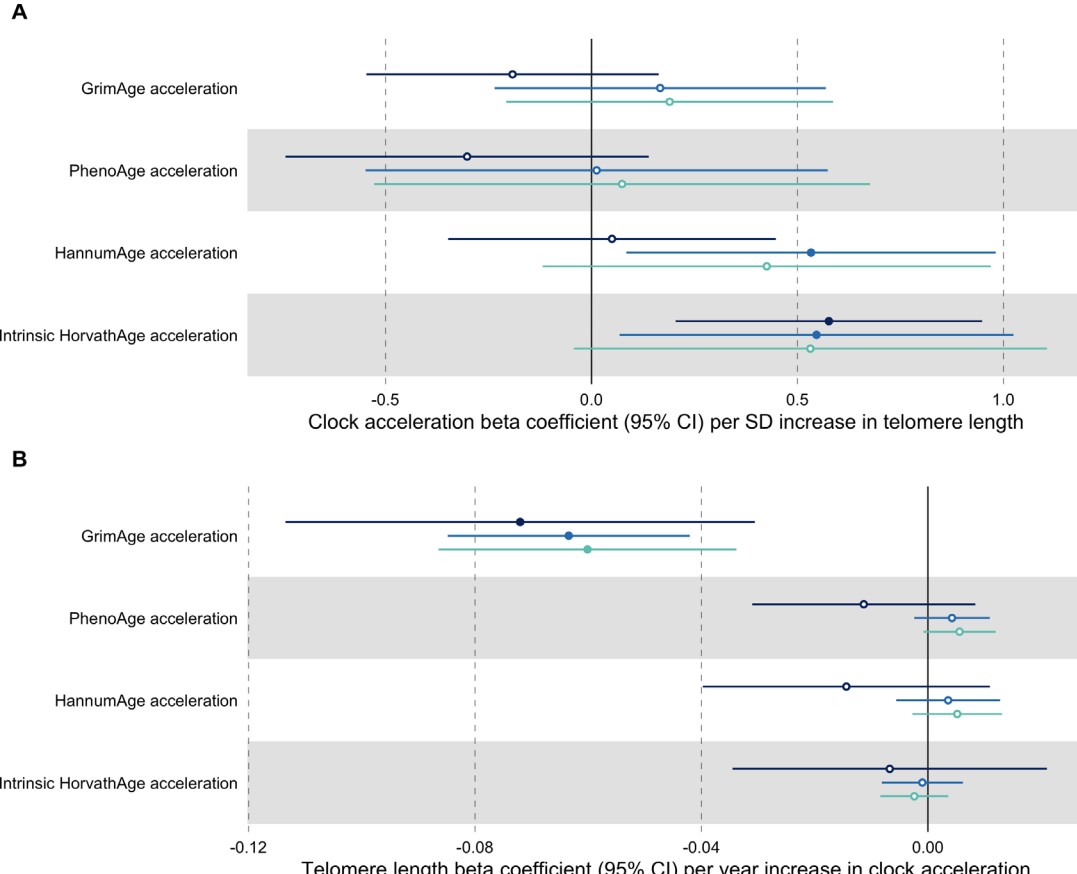

**Appendix 2—figure 13.** Bidirectional Mendelian randomization estimates for epigenetic age acceleration and measured telomere length. Beta coefficients and 95% confidence intervals are reported per (**A**) one standard deviation increase in telomere length and (**B**) 1 year increase in epigenetic clock acceleration. Results were obtained using inverse variance weighted MR (dark blue), weighted median (sky blue) and weighted mode (turquoise) methods. Telomere length, GrimAge, PhenoAge, HannumAge and Intrinsic HorvathAge acceleration were instrumented by 128, 4, 11, 9 and 23 genetic variants, respectively.

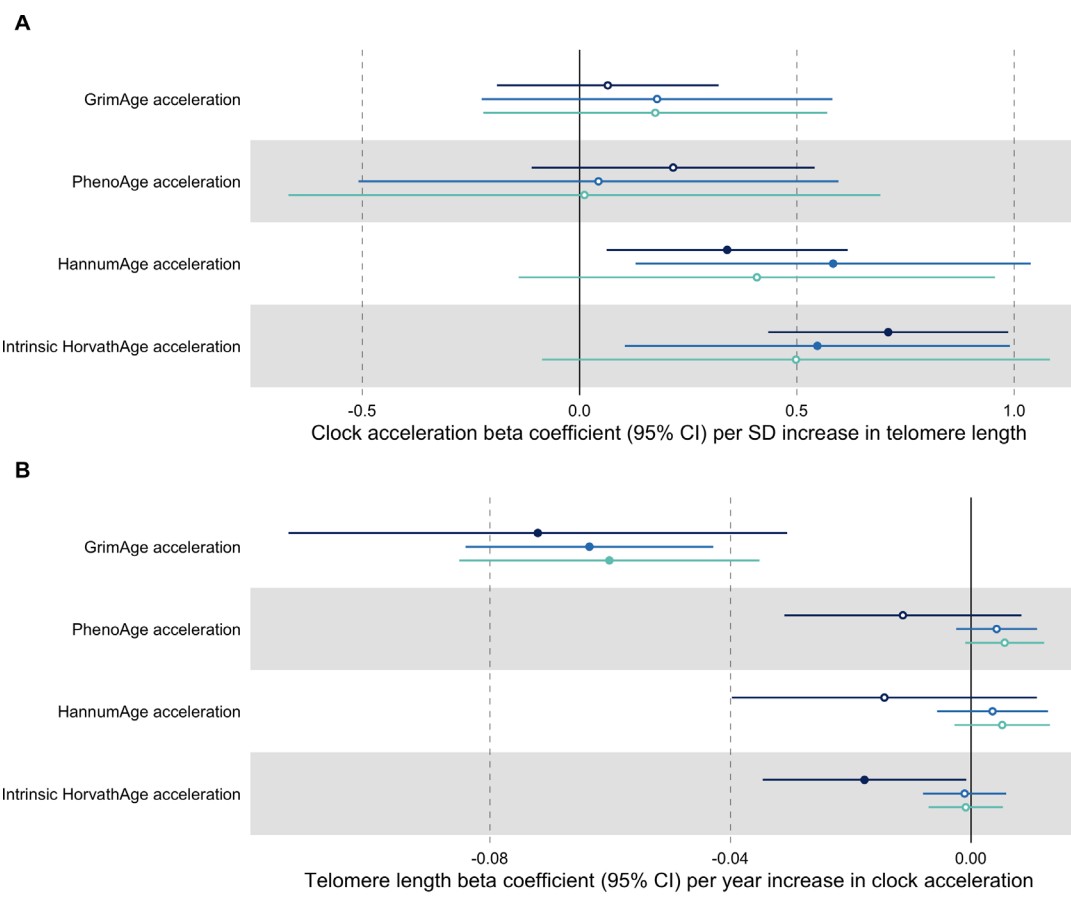

Appendix 2—figure 14. Bidirectional Mendelian randomization estimates for epigenetic age acceleration and measured telomere length after Steiger filtering. Beta coefficients and 95% confidence intervals are reported per (**A**) one standard deviation increase in telomere length and (**B**) 1 year increase in epigenetic clock acceleration. Results were obtained using inverse variance weighted MR (dark blue), weighted median (sky blue) and weighted mode (turquoise) methods. GrimAge, PhenoAge, HannumAge and Intrinsic HorvathAge acceleration were instrumented by 4, 11, 9 and 22 genetic variants, respectively. Telomere length was instrumented by 104, 105, 105 and 109 genetic variants in GrimAge, PhenoAge, HannumAge and Intrinsic HorvathAge acceleration analyses, respectively,.

