## [Editor Report]

This paper is of broad interest to researchers seeking to disentangle the health impact of epigenetic age acceleration, and will provide a substantive empirical contribution to the literature. The authors were very meticulous in addressing all the concerns from the reviewers, which has further improved the paper.

---

## [Decision Letter]

**Decision letter after peer review:**

Thank you for submitting your article "Assessing the causal role of epigenetic clocks in the development of multiple cancers: a Mendelian randomization study" for consideration by *eLife*. Your article has been reviewed by 3 peer reviewers, and the evaluation has been overseen by a Reviewing Editor and a Senior Editor. The following individual involved in review of your submission has agreed to reveal their identity: Daniel W Belsky (Reviewer #2).

As is customary in *eLife*, the reviewers have discussed their critiques with one another. What follows below is the Reviewing Editor's edited compilation of the essential and ancillary points provided by reviewers in their critiques and in their interaction post-review. Please submit a revised version that addresses these concerns directly. Although we expect that you will address these comments in your response letter, we also need to see the corresponding revision clearly marked in the text of the manuscript. Some of the reviewers' comments may seem to be simple queries or challenges that do not prompt revisions to the text. Please keep in mind, however, that readers may have the same perspective as the reviewers. Therefore, it is essential that you attempt to amend or expand the text to clarify the narrative accordingly.

Essential revisions:*Reviewer #1 (Recommendations for the authors):*

This manuscript applies two-sample Mendelian randomization to investigate potential causal effects of epigenetic age acceleration on the risk of multiple cancers. I had some concerns about the study design, the data analysis, as well as the presentation and interpretation of results (see the comments below).

Major comments

1. The authors should assess the validity of the independence assumption (e.g., report the associations of plausible confounders with both the genetic instruments and the outcome) and discuss its impact on the interpretation of results.

2. The authors should present analytical details on the calculation of the genetic variant-exposure and genetic variant-outcome associations, such as the statistical methods, software, and covariates used to estimate these associations. These can help readers to assess the potential bias underlying this study (e.g., a violation of the independence assumption due to population stratification). Any transformation made in the quantitative variables (e.g., epigenetic age acceleration) should also be described, because this can affect the interpretation of the results.

3. Instrument selection: The quality control parameters for the selection of genetic instruments should be provided, including the imputation info score, call rate, minor allele frequency (MAF), and P value from the Hardy-Weinberg equilibrium test, etc. The software used for linkage disequilibrium (LD) clumping and pruning should be provided.

4. Lines 146-148: How were the proportions of trait variance explained by genetic instruments calculated?

5. It is not entirely clear about how multiple testing was addressed. The authors should state whether the correction was for the total number of independent statistical tests (4 epigenetic age acceleration × 5 cancers = 20).

6. Supplementary Table 4: This table presents Mendelian randomization results for "pan cancer inc C44" and "pan cancer". However, the definitions of these outcomes and the statistical methods used to calculate the genetic variant-outcome associations were not provided. "Lung cancer associations were estimated twice, once adjusting for genotyping chip and once without": which set of associations were used for the Mendelian randomization analysis?

7. It would be better to report the association between each genetic instrument and the outcome (i.e., breast, prostate, colorectal, ovarian, and lung cancer).

8. The clinical (or public health) utility of the results can be discussed in more detail.

*Reviewer #2 (Recommendations for the authors):*

I would suggest two additional sets of analyses that could help contextualize findings for the clocks:

One potentially useful evaluation of the authors results would be to conduct a set of analyses of an additional DNAm variable: predicted Telomere Length (TL). There is strong evidence from prior MR studies that shorter TL is protective of risk for several cancers. It would build confidence in the clock results reported here if they were to find similar evidence for DNAm-predicted TL based on GWAS in the same sample used for GWAS of the DNAm clocks.

A second analysis worth conducting would be to focus on SNPs identified in GWAS of familial longevity.

Together, results from these analyses would speak to whether the clock results are similar to or different from results for genetics identified by other markers of biological aging.

*Reviewer #3 (Recommendations for the authors):*

Results:

– For positive findings (e.g., GrimAge and CRC), the scatter plot of SNP associations for G-X vs G-Y should be presented as a primary figure, so readers can have a more complete picture of the result.

– In the supp figures, why do the authors not consistently report the association estimates for the increased (or decreased) acceleration allele? Currently, it looks like associations for 3 SNPs are reported in terms of the decreased acceleration allele, but one is reported in terms of the increased acceleration allele.

– for CAUSE results in the supp figures have too many components which will not be easily interpretable by readers. Either simplify or provide more explanation.

– " … results replicated when using UK Biobank data alone (IVW OR=1.15, 95%CI 1.04-1.28, p=0.007) (Figure 2, Supplementary Table 4)." I do not think this result can be called replication, because the GECCO data contains the UKB cases (correct?).

– can you briefly mention how the random effects IVW approach differs from the standard IVW approach? And multiplicative refers to what? The way the SNP effects are combined to generate the IVW estimate?

Discussion

– The challenges associated with additional replication of these findings should be mentioned. These results are based on very large sample sizes. Similar datasets for replication purposes may not be available in the short term.

– when describing the prior literature, make very clear which studies used MR and which used measured DNA methylation.

– The most convincing association reported is between GrimAge and colon cancer. GrimAge was trained on smoking, a risk factor for colon cancer. is it possible that the GrimAge IVs somehow reflect DNA methylation features related to smoking? Or would these IVs be more related to the "intrinsic aging" dimensions of the GrimAge clock (given that smoking occurs after assignment of genotype at birth)? I'm not sure exactly how to interpret.

– would the authors consider the lung findings to be consistent with what is known regarding telomere length and lung cancer risk?

– " … their effect estimates are not directly comparable to those obtained in the main analyses due to cases in the GWAS-by-proxy of parental endpoints being defined as either or both parents reportedly having a type of cancer." This is in the methods but probably belongs in the discussion as a limitation. These results should probably not be referred to as "replication" since the effect estimates cannot be compared. "consistency" may be more appropriate language.

---

## [Author Response]

Essential revisions:Reviewer #1 (Recommendations for the authors):This manuscript applies two-sample Mendelian randomization to investigate potential causal effects of epigenetic age acceleration on the risk of multiple cancers. I had some concerns about the study design, the data analysis, as well as the presentation and interpretation of results (see the comments below).1. The authors should assess the validity of the independence assumption (e.g., report the associations of plausible confounders with both the genetic instruments and the outcome) and discuss its impact on the interpretation of results.

Confounding by population stratification and ancestry is unlikely as the cancer GWAS analysed have been adjusted for principal components and restricted to participants of European descent. Nevertheless, we have now conducted additional MR analyses using negative control outcomes (i.e., skin colour, ease of skin tanning). We would not anticipate observing evidence of causality between epigenetic clocks and these outcomes. Any evidence of an effect would be indicative of confounding (i.e., a common cause of both genetic variants and cancer) (Sanderson et al., 2021, available at: https://academic.oup.com/ije/article/50/4/1350/6133127?login=true). This was performed to robustly assess whether population stratification has been fully accounted for through adjustments in the GWAS. The results suggest that our MR analyses are unlikely to be affected by population stratification.

We also conducted MR analyses to assess the genetically predicted effect of epigenetic age acceleration on potential confounders of the exposure-outcome association to detect potential violations of the exclusion restriction assumption (i.e., genetic variants should not affect the outcome via pleiotropic pathways, they should only affect the outcome via their effect on the exposure). From now on, we will refer to these confounders as cancer risk factors. We investigated the following cancer risk factors: body mass index, waist circumference, pack years of smoking, time spent doing vigorous physical activity, age completed full time education, years of schooling, and alcohol intake frequency. The results suggest that there is little evidence of horizontal pleiotropy via the cancer risk factors investigated.

Taken together, these additional analyses suggest that the genetic instruments used in our study are not related to potential confounders and so are unlikely to violate the independence and exclusion restriction assumptions.

We have included the following paragraph in the Methods section (pages 12–13, lines 263–274):

“We further assessed the validity of the independence assumption by conducting MR analyses using negative control outcomes (i.e., skin colour, ease of skin tanning). Evidence of causality between our genetic instruments for epigenetic age acceleration and these negative control outcomes would suggest potential bias due to population stratification that has not been fully accounted for through adjustments in the GWAS. We also assessed the genetically predicted effect of epigenetic age acceleration on cancer risk factors (i.e., body mass index, waist circumference, pack years of smoking, time spent doing vigorous physical activity, age completed full time education, years of schooling, and alcohol intake frequency) to detect potential violations of the exclusion restriction assumption. GWAS data for negative control outcomes and cancer risk factors were obtained using the University of Bristol’s IEU OpenGWAS API (for more details, see Appendix 1).”

We also mention these analyses in the Results and Discussion sections. Both figures have been included as additional figures in Appendix 2 of the revised manuscript. We have also included negative control outcome and cancer risk factor results in Supplementary File 1.

2. The authors should present analytical details on the calculation of the genetic variant-exposure and genetic variant-outcome associations, such as the statistical methods, software, and covariates used to estimate these associations. These can help readers to assess the potential bias underlying this study (e.g., a violation of the independence assumption due to population stratification). Any transformation made in the quantitative variables (e.g., epigenetic age acceleration) should also be described, because this can affect the interpretation of the results.

Thank you for these suggestions to help improve the clarity of our Methods section for readers. GWAS data used for the associations between genetic variants and exposures have been briefly described in the Methods section under “Genetic instruments for epigenetic age acceleration” (pages 8–9, lines 140–179). We have included extra information regarding statistical methods, software and covariates used to estimate these associations:

“We obtained summary genetic association estimates for epigenetic age acceleration measures of HannumAge, Intrinsic HorvathAge, PhenoAge and GrimAge from a recent GWAS meta-analysis of biological ageing, which included 34,710 participants of European ancestry. Across the 28 European ancestry studies considered in the analysis, 57.3% of participants were female. A detailed description of the methods that were used can be found in the publication by McCartney et al., In short, the Horvath epigenetic age calculator software (https://dnamage.genetics.ucla.edu) or standalone scripts were used to calculate age adjusted DNAm estimates. Outlier samples with clock methylation estimates of +/− 5 s.d. from the mean were excluded from further analysis. SNPs were genotyped and imputed independently for each cohort included in the meta-analysis. Genotypes were imputed using either the HRC or the 1000 Genomes Project Phase 3 reference panels in all cohorts but the Sister Study (which did not have imputed data at the time of analysis) and the Genetics of Lipid Lowering Drugs and Diet Network Study (which used whole-genome sequencing data). GWAS summary statistics were obtained in each cohort using additive linear models adjusted for sex and genetic principal components, and they were later processed and harmonised using the ‘EasyQC’ R package. Fixed effect meta-analyses were performed using the METAL software.”

GWAS summary statistics used to determine associations between genetic variants and outcomes have been briefly described in Appendix 1. We have presented information regarding the statistical methods, software, covariates used in the UK Biobank and FinnGen. We have added the information that was missing for the international cancer genetic consortia.

We did not apply transformations to the quantitative variables, as genetic association data for the four measures of epigenetic ageing obtained from McCartney et al., had already been reported in years (our results are reported per year increase in epigenetic age acceleration). We clarified this in the manuscript as follows (page 15, lines 334–336):

“MR results were reported as the odds ratio (OR) of site-specific cancer per one year increase in genetically predicted epigenetic age acceleration. These did not require any scale transformations, as the GWAS of biological ageing reported epigenetic age acceleration in years”.

3. Instrument selection: The quality control parameters for the selection of genetic instruments should be provided, including the imputation info score, call rate, minor allele frequency (MAF), and P value from the Hardy-Weinberg equilibrium test, etc. The software used for linkage disequilibrium (LD) clumping and pruning should be provided.

We have edited the paragraph regarding the selection of genetic instruments as follows (page 9, line 164):

“We used the clump_data function in the ‘TwoSampleMR’ R package to select GWAS-significant SNPs (p<5×10−8) for each epigenetic age acceleration measure and perform linkage disequilibrium (LD) clumping (r2<0.001) using the European reference panel from the 1000 Genomes Project Phase 3 v5.”

Information regarding the quality control parameters used in the GWAS are available in McCartney et al., (Additional file 1). Because McCartney et al., is a meta-analysis of several GWAS that used different quality control parameters, it is not possible to include all the details in the present manuscript. However, we have pointed readers who wish to know more information about these to the publication by McCartney et al., (page 8, line 147):

“A detailed description of the methods that were used can be found in the publication by McCartney et al.”

4. Lines 146-148: How were the proportions of trait variance explained by genetic instruments calculated?

We have added the following explanation in the text (page 9, lines 170–175):

“The proportions of trait variance explained by genetic instruments (R^2^) and instrument strength (F-statistic) were calculated using the following formulae: R^2^ = (2β^2^×MAF×(1-MAF))/(2β^2^×MAF×(1-MAF)+2N×MAF×(1-MAF)×SE^2^) and F = (R^2^×(N-2))/(1-R^2^) (where MAF = effect allele frequency, β = effect estimate of the SNP in the exposure GWAS, SE = standard error, N = sample size)”.

5. It is not entirely clear about how multiple testing was addressed. The authors should state whether the correction was for the total number of independent statistical tests (4 epigenetic age acceleration × 5 cancers = 20).

Initially, we had applied multiple testing corrections to the pooled main IVW results separately for each clock. Specifically, a total of 5 independent statistical tests were considered for each clock, i.e., 1 clock x 5 cancers = 5 tests. However, after this reviewer’s suggestion we have decided to apply a more stringent FDR correction to our pooled IVW MR p-values, considering a total of 20 independent statistical tests (4 clocks x 5 cancers).

We have added the following in the Methods section (page 12, lines 244–245):

“A Benjamini-Hochberg false discovery rate (FDR) <5% was used to correct the pooled main IVW results for multiple testing. This correction was applied considering a total of 20 independent statistical tests (4 clocks x 5 cancers = 20).”.

FDR p-values have been updated accordingly in Supplementary File 1—Table 6 and in the Results section of the manuscript. Also, a footnote has been included in Supplementary File 1—Table 6:

“FDR corrections considered a total of 20 independent statistical tests (5 cancers x 4 clocks = 20)”.

It is important to note that after changing the number of tests included in the FDR correction, our results for GrimAge acceleration and prostate cancer no longer survive the multiple testing correction. We have therefore edited the manuscript accordingly (please see the tracked changes in the Results and Discussion sections). The lack of a robust association between GrimAge acceleration and prostate cancer in the main analysis is consistent with LD score regression results for GrimAge and prostate cancer (i.e., little evidence of genetic correlation), with MR results for GrimAge and parental history of prostate cancer (i.e., little evidence of a genetically predicted effect), and with additional analyses that were conducted in response to comments 7 (reviewer 1) and 1 (reviewer 3) below.

In contrast, our results for GrimAge and colorectal cancer withstood the new FDR correction threshold accounting for 20 independent statistical tests. We decided to focus on these findings throughout the manuscript. For instance, we adapted Figure 5 so that it only shows colorectal cancer subtypes.

6. Supplementary Table 4: This table presents Mendelian randomization results for "pan cancer inc C44" and "pan cancer". However, the definitions of these outcomes and the statistical methods used to calculate the genetic variant-outcome associations were not provided. "Lung cancer associations were estimated twice, once adjusting for genotyping chip and once without": which set of associations were used for the Mendelian randomization analysis?

We have removed UK Biobank pan cancer results (“pan cancer inc C44” and “pan cancer”) from the table, as these were not meant to be included.

Mendelian randomization analyses were conducted using both adjusted and unadjusted lung cancer associations, but only unadjusted findings were included in the cross-cohort meta-analysis. It has been recommended that MR analyses be conducted using GWAS data without genotyping array adjustment for outcomes that are related to smoking, as this may induce collider bias into findings. However, both adjusted and unadjusted findings are presented in Supplementary File 1—Table 4 to facilitate their comparison. In this instance, the results were almost identical, so it is unlikely that the use of unadjusted lung cancer GWAS estimates in the meta-analysis biased our results. We have added the following sentence in Appendix 1 to clarify this:

“Since UKBiLEVE participants were genotyped using a different array and using adjusted lung cancer estimates may introduce collider bias, we only included MR results obtained using the unadjusted lung cancer GWAS estimates in the meta-analysis”.

We have also added the following sentence as a footnote in Supplementary File 1—Table 4 to clarify this:

“We only included MR results obtained using the unadjusted lung cancer GWAS estimates in the meta-analysis (UKBiLEVE participants were genotyped using a different array, so using adjusted lung cancer estimates may introduce collider bias), but both unadjusted and adjusted MR results have been presented in Supplementary File 1—Table 4 to facilitate their comparison”.

7. It would be better to report the association between each genetic instrument and the outcome (i.e., breast, prostate, colorectal, ovarian, and lung cancer).

We agree with this reviewer’s suggestion. We have conducted additional analyses in response to this comment, as we had originally conducted a meta-analysis of cohort-specific MR estimates, as opposed to a meta-analysis of SNP effects followed by MR analyses. We have now estimated pooled SNP effects using the METAL software and then performed two-sample MR analyses using these genetic variants. The methodology used has been described in the Methods section and the results have been included in Supplementary File 1—Table 9.

8. The clinical (or public health) utility of the results can be discussed in more detail.

We have included the following paragraph in the Discussion section (pages 34–35, lines 909–923):

“From a public health perspective, our work provides potentially relevant findings. Observational and Mendelian randomization studies suggest that GrimAge acceleration may be influenced by several cancer risk factors, such as obesity and smoking. If GrimAge acceleration is a causal mediator between these risk factors and colorectal cancer, the GrimAge clock may be a treatable intermediary when targeting the underlying risk factors is not feasible or too difficult to accomplish. It could also be targeted in populations at high-risk of colorectal cancer. Nevertheless, we think it may be too early to make claims regarding the clinical utility of our findings. The GrimAge clock has only recently been created and very few studies have assessed its association with colorectal cancer. More research is required to corroborate our findings and to evaluate whether GrimAge acceleration can be modified through lifestyle or clinical interventions.”

Reviewer #2 (Recommendations for the authors):I would suggest two additional sets of analyses that could help contextualize findings for the clocks:One potentially useful evaluation of the authors results would be to conduct a set of analyses of an additional DNAm variable: predicted Telomere Length (TL). There is strong evidence from prior MR studies that shorter TL is protective of risk for several cancers. It would build confidence in the clock results reported here if they were to find similar evidence for DNAm-predicted TL based on GWAS in the same sample used for GWAS of the DNAm clocks.

Thank you for your suggestions to help refine this manuscript. Unfortunately, a GWAS of DNAm-predicted telomere length has not been conducted using the same sample used in McCartney et al., (available at: https://genomebiology.biomedcentral.com/articles/10.1186/s13059-021-02398-9). We agree that it would be interesting to compare our results with those from an MR of DNAm telomere length and cancer, especially since DNAm telomere length has been more strongly associated with age than measured telomere length (Lu et al., 2019 available at: https://www.ncbi.nlm.nih.gov/pmc/articles/PMC6738410/). Although it is not possible to run these analyses now, we will consider running them in the future when the data become available.

A recent large-scale GWAS identified novel genetic variants associated with measured telomere length (Codd et al., 2021 available at: https://www.nature.com/articles/s41588-021-00944-6, data available through MR-Base under id “ieu-b-4879”). We conducted additional bidirectional MR analyses for GrimAge acceleration and telomere length to try to get a better understanding of the relationship between these two measures of biological ageing (see Author response image 1). The results suggest that GrimAge acceleration may be a cause of telomere shortening. This is logical, as telomeres shorten with age (and GrimAge acceleration is a measure of biological ageing).

**Author response image 1. sa2fig1:** Bidirectional Mendelian randomization estimates for GrimAge acceleration and telomere length. Here we see the genetically predicted effect of (A) telomere length on GrimAge acceleration and (B) GrimAge acceleration on telomere length.

According to Haycock et al.’s MR analyses (available at: https://www.ncbi.nlm.nih.gov/pmc/articles/PMC5638008/), genetically predicted longer telomere length may increase the risk of lung cancer (OR 1.71 per 1-SD increase in telomere length, 95%CI 1.44-2.04). In contrast, they find little evidence of causality between telomere length and breast (OR 1.08, 95%CI 0.99-1.19), ovarian (OR 1.09, 95%CI 0.94-1.27), prostate (OR 1.12, 95%CI 0.96-1.30) and colorectal cancer (OR 1.09, 95%CI 0.91-1.31). The paucity of evidence in favour of an association could be due to a lack of statistical power to detect small effects (the MR analyses were powered to detect ORs ≥2.0 per SD increase in telomere length, assuming an α of 0.01). It is important to note that the directionality of the associations is consistent across cancer sites. Also, another MR study suggest that longer telomeres increase the risk of overall cancer (OR 1.11, 95%CI 1.06-1.16) (Kuo et al., 2019, available at: https://onlinelibrary.wiley.com/doi/10.1111/acel.13017).

Taken together, these results suggest that GrimAge acceleration could affect cancer risk via its effect on telomere length (GrimAge acceleration causes shorter telomeres, which decrease the risk of cancer). However, this does not explain why we observe evidence of a positive effect of GrimAge acceleration on colorectal cancer (see Author response image 2) and a null effect of GrimAge acceleration on other cancers (see Author response image 3). One plausible explanation is that GrimAge acceleration may also affect cancer risk via pathways other than those related to cellular division (e.g., inflammatory cytokines, microbiome, glycogen metabolism, enzyme activity). The positive effect of GrimAge acceleration on cancer via these other pathways may counteract the negative effect mediated via telomere length, resulting in null MR results for GrimAge acceleration and breast, ovarian, prostate and lung cancer, and positive MR results for GrimAge acceleration and colorectal cancer. Further research is required to understand the biological pathways underlying our MR results.

**Author response image 2. sa2fig2:** Causal associations between GrimAge acceleration, shorter telomere length and colorectal cancer.

**Author response image 3. sa2fig3:** Causal associations between GrimAge acceleration, shorter telomere length and breast, ovarian, prostate and lung cancer.

We included the following in the Methods section (page 15, lines 344–353):

“Finally, bidirectional MR analyses were conducted to assess the causality and directionality of the link between epigenetic clock acceleration and telomere length, another measure of biological ageing that has been shown to influence cancer risk in prior MR studies. The MR Steiger test of directionality was used to confirm the assumption that the exposure causes the outcome is valid. We also corroborated our findings by rerunning the analyses using data that had undergone Steiger filtering to remove SNPs that explained more variance in the outcome than in the risk factor. Genetic association data for measured telomere length were obtained from Codd et al., the largest GWAS of telomere length available through the OpenGWAS API at the time of analysis (N=472,174, for more details, see Appendix 1)”.

We also added the bidirectional MR results in the Results section (page 23, lines 644–664):

“In bidirectional MR analyses, we found evidence that genetically predicted GrimAge acceleration may be a cause of telomere shortening (IVW β coefficient=-0.07 per year increase in GrimAge acceleration, 95%CI -0.09–-0.05, p<0.001) and that genetically predicted longer telomere length may increase Intrinsic HorvathAge acceleration (IVW β coefficient=0.57 per standard deviation increase in telomere length, 95%CI 0.39–0.77, p=0.002) (Appendix 2—Figure 13, Supplementary File 1—Table 14). Steiger filtering showed that all genetic instruments for GrimAge acceleration were stronger predictors of GrimAge acceleration than telomere length. In contrast, it identified 20 genetic instruments for telomere length that were better predictors of Intrinsic HorvathAge acceleration than telomere length (Supplementary File 1—Table 15). After removing these SNPs from the analyses, the results were still suggestive of an effect of telomere length on Intrinsic HorvathAge acceleration (IVW β coefficient=0.71 per standard deviation increase in telomere length, 95%CI 0.57–0.85, p<0.001) (Appendix 2—Figure 14, Supplementary File 1—Table 16). There was little evidence of causality between other measures of epigenetic age acceleration and telomere length (Appendix 2—Figure 13, Supplementary File 1—Table 14)”.

The following has been incorporated into the Discussion section (pages 32–33, lines 848–863):

“Although little is known about the underlying mechanisms, GrimAge may plausibly influence cancer risk through hormonal, inflammatory and metabolic processes. In bidirectional MR analyses, we found evidence that genetically predicted GrimAge acceleration may be a cause of telomere shortening, another marker of biological ageing. Shorter telomeres have been shown to lower cancer risk in prior MR analyses, so it is plausible that GrimAge acceleration decreases cancer risk, at least in part, via its effect on telomere length. GrimAge acceleration may still increase cancer risk via pathways other than those related to cellular division. The positive effect of GrimAge acceleration on cancer via these other pathways may counteract the negative effects mediated via telomere length, resulting in null MR results for GrimAge acceleration and breast, ovarian, prostate and lung cancer, and positive MR results for GrimAge acceleration and colorectal cancer. To better understand the biology of ageing, future studies should consider running MR analyses using data on DNAm-predicted telomere length, since DNAm telomere length is independent of telomerase activity and has been more strongly associated with age than measured telomere length.”

A second analysis worth conducting would be to focus on SNPs identified in GWAS of familial longevity.Together, results from these analyses would speak to whether the clock results are similar to or different from results for genetics identified by other markers of biological aging.

We have conducted the requested MR analyses using UK Biobank GWAS data for parental longevity (combined parental age at death) obtained through the OpenGWAS API (dataset id: ebi-a-GCST006702, PMID: 29227965). We ran the analyses for all cancers using GWAS data from FinnGen, the UK Biobank and International cancer genetic consortia (except for GECCO, since we do not have complete summary statistics for colorectal cancer from this data source). The results suggest that genetic liability to increased longevity does not influence cancer risk (see Author response image 4). We also performed bidirectional MR analyses to investigate the relationship between epigenetic age acceleration and longevity and found no evidence of a causal effect either way: GrimAge acceleration does not seem to be a cause of longevity and longevity does not seem to be a cause of GrimAge acceleration (see Author response image 5).

**Author response image 4. sa2fig4:** Inverse-variance weighted Mendelian randomization estimates for genetically predicted effects of parental longevity (combined parental age at death) on multiple cancers.

**Author response image 5. sa2fig5:** Bidirectional Mendelian randomization estimates for GrimAge acceleration and parental longevity. Here we see the genetically predicted effect of (A) parental longevity estimated as combined parental age at death on GrimAge acceleration and (B) GrimAge acceleration on parental longevity estimated as combined parental age at death.

We would prefer not to include these additional analyses in the manuscript because we feel it is challenging to interpret the findings of an MR study assessing the genetically predicted effect of parental longevity on cancer risk, especially given the potential for survival bias and dynastic effects in this context. We would be happy to include them at the reviewer/editor’s discretion.

Reviewer #3 (Recommendations for the authors):Results:– For positive findings (e.g., GrimAge and CRC), the scatter plot of SNP associations for G-X vs G-Y should be presented as a primary figure, so readers can have a more complete picture of the result.

This point relates to reviewer 1’s seventh comment. We have now conducted a GWAS meta-analysis of individual SNP effects from the cancer genetic association data and used these summary statistics to perform two-sample MR analyses. We then created scatter plots to depict the associations requested. The scatter plot for GrimAge and colorectal cancer has been included as a primary figure: Figure 4. Scatter plots for other associations are now available in Appendix 2.

– In the supp figures, why do the authors not consistently report the association estimates for the increased (or decreased) acceleration allele? Currently, it looks like associations for 3 SNPs are reported in terms of the decreased acceleration allele, but one is reported in terms of the increased acceleration allele.

Thank you for noting this. We have changed the sign of the negative betas to positive and flipped the corresponding alleles accordingly. Now all the association estimates are reported in terms of the increased acceleration allele.

– for CAUSE results in the supp figures have too many components which will not be easily interpretable by readers. Either simplify or provide more explanation.

We agree. We have simplified the results and provided more explanation. This relates to reviewer 1’s fourth minor comment.

We have re-labelled the X- and Y-axes. We also simplified the CAUSE output. We have additionally provided a detailed description of the CAUSE output as follows:

“The ELPD Contribution plot shows the relative contribution of each SNP to the CAUSE test statistic. Only SNPs with p<5e-8 are shown. SNPs represented by larger circles reflect smaller p-values for the associations between genetic variants and GrimAge acceleration. SNPs that contribute more to the causal model are shown in warmer tones (i.e., red), while those that contribute more to the sharing model are shown in colder tones (i.e., blue). The δ_elpd is the statistic used to compare models. It is equal to elpd(model 1)- elpd(model 2). In the upper table, positive δ_elpd’s suggest that model 1 is a better fit to the data than model 2 (i.e., that the null model is better than the sharing model in row 1, that the null model is better than the causal model in row 2, and that the sharing model is better than the causal model in row 3). The corresponding p-values test whether model 2 is a better fit than model 1. Here, row 3 suggests that the causal model is not a better fit than the sharing model (the δ_elpd is positive and the p-value is 1, so there is no detectable evidence against the null hypothesis that the sharing model is better than the causal model). In the bottom table, eta represents the sharing factor effect (SNPs affect shared factor and shared factor simultaneously affects GrimAge and colorectal cancer) and γ represents the causal factor effect (SNPs affect GrimAge and GrimAge affects colorectal cancer). Here, “0 (-0.04, 0.04)” represents the genetically predicted effect of GrimAge acceleration on colorectal cancer after adjusting for correlated and uncorrelated horizontal pleiotropy (results in log odds ratio scale).”

– " … results replicated when using UK Biobank data alone (IVW OR=1.15, 95%CI 1.04-1.28, p=0.007) (Figure 2, Supplementary Table 4)." I do not think this result can be called replication, because the GECCO data contains the UKB cases (correct?).

We agree and have edited the text accordingly (page 21, line 526). We replaced “results replicated when using UK Biobank data alone” with “results were consistent when using UK Biobank data alone”.

– can you briefly mention how the random effects IVW approach differs from the standard IVW approach? And multiplicative refers to what? The way the SNP effects are combined to generate the IVW estimate?

The multiplicative random effects IVW method is the default in the ‘TwoSampleMR’ package. It has been recommended as the default because it accounts for excess heterogeneity across SNP-specific estimates (as opposed to the fixed effect IVW method) and it does not affect the relative weights of individual SNP estimates (in contrast to the additive random effects IVW method).

Fixed effect IVW MR assigns all SNPs the same weight, assuming that all genetic variants are valid instruments, that they affect the exposure via the same biological pathways, and that their effects on the outcome are of equal magnitude. It assumes homogeneity, which is rarely the case and the reason why it is preferable to use random effects IVW approaches that account for excess heterogeneity.

Additive and multiplicative random effects produce the same results in presence of balanced pleiotropy. In cases of directional pleiotropy, the additive method is less robust, as it assigns a disproportionately high weight to outlying estimates which are likely due to pleiotropy.

Fixed effect and multiplicative random effects IVW MR produce the same effect estimates, but multiplicative random effects produce wider confidence intervals to account for heterogeneity. Therefore, we have decided to use the default, multiplicative random effects method because it combines individual SNP ratio estimates in an IVW meta-analysis that accounts for heterogeneity.

We have included the following in the Methods section of the manuscript (page 11, lines 232–236):

“This is the default IVW MR method in the "TwoSampleMR” R package, as it accounts for excess heterogeneity across SNP-specific estimates (as opposed to the fixed effect IVW method) and it does not affect the relative weights of individual SNP estimates (in contrast to the additive random effects IVW method)”.

References:

Burgess S, Davey Smith G, Davies NM et al., Guidelines for performing Mendelian randomization investigations [version 2; peer review: 2 approved]. Wellcome Open Res 2020, 4:186 (https://doi.org/10.12688/wellcomeopenres.15555.2)

Bowden J, Del Greco M F, Minelli C, Davey Smith G, Sheehan N, Thompson J. A framework for the investigation of pleiotropy in two-sample summary data Mendelian randomization. Stat Med. 2017;36(11):1783-1802. doi:10.1002/sim.7221

Discussion– The challenges associated with additional replication of these findings should be mentioned. These results are based on very large sample sizes. Similar datasets for replication purposes may not be available in the short term.

We agree that this is a good point which should be acknowledges in the manuscript. We have therefore added the following in the Discussion section (page 33, lines 884–886):

“In spite of these suggestions, we acknowledge that it may be challenging to get access to suitable datasets for replication purposes in the short term.”.

– when describing the prior literature, make very clear which studies used MR and which used measured DNA methylation.

We have made some edits to the Discussion section. Hopefully, it is now clear that all previous literature used measured DNA methylation data. We added the text in bold to clarify the following sentence (page 28, lines 665–666):

“However, our findings contrast with those highlighted in Hillary et al., an observational cohort study that used Generation Scotland data.”.

To date, McCartney et al., (the GWAS used in our analyses) is the only study that has used MR to explore the genetically predicted effect of epigenetic age acceleration on cancer. This was explained in the Introduction section (page 7, lines 119–123):

“McCartney et al.,^20^ used IVW MR, MR-Egger, weighted median and weighted mode methods to explore the genetically predicted effects of HannumAge, Intrinsic HorvathAge, PhenoAge and GrimAge acceleration on breast, ovarian and lung cancer. Here we extend this analysis to include colorectal and prostate cancer (two of the most common cancers worldwide^21^) and use additional methods and datasets to verify the robustness of our findings.”.

We also reiterated this in the Discussion section (note that we added ‘MR’ in the sentence to make it clearer) (page 29, lines 734–736):

“For instance, we pooled results from multiple sources using fixed effect meta-analysis to improve the precision of the MR estimates presented in McCartney et al^20^.”.

– The most convincing association reported is between GrimAge and colon cancer. GrimAge was trained on smoking, a risk factor for colon cancer. is it possible that the GrimAge IVs somehow reflect DNA methylation features related to smoking? Or would these IVs be more related to the "intrinsic aging" dimensions of the GrimAge clock (given that smoking occurs after assignment of genotype at birth)? I'm not sure exactly how to interpret.

This is a good question. Although it is plausible that GrimAge acceleration captures features related to smoking (as it was trained on DNAm pack-years), we do not think that the four SNPs used as genetic instruments for GrimAge acceleration capture these signals. Linkage disequilibrium score regression results show that GrimAge acceleration is correlated with lung cancer (which is known to be associated with smoking). Nevertheless, we do not find MR evidence to suggest that genetically predicted GrimAge acceleration increases the risk of lung cancer. Therefore, we believe GrimAge acceleration may increase the risk of colorectal cancer via an alternative pathway to smoking.

We have conducted additional MR analyses in response to reviewer 1’s first comment. Here, we assessed whether genetically predicted GrimAge acceleration influences pack-years of smoking. We did not find evidence of an effect. This suggests that the genetically predicted association between GrimAge acceleration and colorectal cancer is unlikely to be mediated by smoking, corroborating our speculations.

– would the authors consider the lung findings to be consistent with what is known regarding telomere length and lung cancer risk?

According to Haycock et al., (available at: https://www.ncbi.nlm.nih.gov/pmc/articles/PMC5638008/), 1-SD increase in genetically predicted telomere length may increase the risk of lung cancer (OR 1.71, 95%CI 1.44-2.04), specifically lung adenocarcinoma (OR 3.19, 95%CI 2.40-4.22). Although our linkage disequilibrium score (LDSC) results show evidence of a positive genetic correlation between GrimAge acceleration and lung cancer, our MR results suggest that the effect is not causal. Interestingly, our results for GrimAge acceleration and colorectal cancer also contrast with those from the telomere length study. We observe evidence of a positive genetic correlation and a positive causal effect between GrimAge acceleration and colorectal cancer, while Haycock et al., shows little evidence of an association between telomere length and colorectal cancer (OR 1.09 95%CI 0.91-1.31) (available at https://www.ncbi.nlm.nih.gov/pmc/articles/PMC5638008/). We were surprised by our findings, as we would have expected to find results consistent with those based on telomere length, another biomarker of biological ageing.

One possible explanation for this discrepancy could be that genetically longer telomere length is associated with an increase in somatic mutations (which is consistent with longer telomeres being associated with a greater number of lifetime cell divisions and cancer risk), while GrimAge acceleration mechanisms may have nothing to do with cell division. In fact, observational and GWAS evidence suggests that epigenetic ageing is independent of telomerase activity and replicative senescence-mediated ageing (Kabacik et al., 2018, available at: https://www.aging-us.com/article/101588/text).

In response to Reviewer 2’s first comment, we have conducted additional bidirectional MR analyses for GrimAge acceleration and telomere length to try to get a better understanding of the relationship that exists between these two measures of biological ageing (see Author response image 1). The results suggest that genetically predicted GrimAge acceleration may be a cause of telomere shortening. This is logical, as telomeres shorten with age (and GrimAge acceleration is a measure of biological ageing).

Taken together, these results suggest that GrimAge acceleration may decrease cancer risk via its effect on telomere length (GrimAge acceleration causes shorter telomeres, which decrease the risk of cancer), and simultaneously increase cancer risk via pathways other than those related to cellular division (e.g., inflammatory cytokines, microbiome, glycogen metabolism, enzyme activity). The positive effect of GrimAge acceleration on cancer via these other pathways may counteract the negative effect mediated via telomere length, resulting in null MR results for GrimAge acceleration and breast, ovarian, prostate and lung cancer (see Author response image 3), and positive MR results for GrimAge acceleration and colorectal cancer (see Author response image 4). Further research is required to understand the biological pathways underlying our MR results.

– " … their effect estimates are not directly comparable to those obtained in the main analyses due to cases in the GWAS-by-proxy of parental endpoints being defined as either or both parents reportedly having a type of cancer." This is in the methods but probably belongs in the discussion as a limitation. These results should probably not be referred to as "replication" since the effect estimates cannot be compared. "consistency" may be more appropriate language.

We agree. We have moved the text to the Discussion section (page 33, lines 877–880):

“Additionally, our analyses could be replicated using other large independent cancer datasets. Although we conducted MR analyses on parental history of cancer, their effect estimates are not directly comparable to those obtained in the main analyses due to cases in the GWAS-by-proxy of parental endpoints being defined as either or both parents reportedly having a type of cancer”.

In the Discussion section (page 30, line 805), we have replaced “…sought to replicate our results using UK Biobank GWAS data on parental history of cancer and LD Score regression” with “…sought to corroborate our results using UK Biobank GWAS data on parental history of cancer and LD Score regression”.